# Evidence for preferred propagating terrestrial heatwave pathways due to Rossby wave activity

Mingzhao Wang[1,2,9], Yu Huang [2,3,9] ✉, Christian L. E. Franzke [4,5], Naiming Yuan [6,7,8] ✉, Zuntao Fu [1] ✉ & Niklas Boers [2,3]

Terrestrial heatwaves are prolonged hot weather events often resulting in widespread socioeconomic impacts. Predicting heatwaves remains challenging, partly due to limited understanding of the events' spatial evolution and underlying mechanisms. Heatwaves were mainly examined at fixed stations, with little attention given to the fact that the center of a heatwave can move a long distance. Here, we examine the spatial propagation of terrestrial heatwaves using a complex network algorithm, and find four preferred propagation pathways of terrestrial heatwaves in the northern hemisphere. Along each preferred pathway, heatwaves evolve in two ways: propagating along the pathway or being stationary. We show that the propagating heatwave pathways are consistent with the movement of Rossby wave trains, and that both are guided by enhanced Rossby wave flux activities. The detected propagation pathways are found to provide prior knowledge for occurrences of downstream heatwaves that can be used for identifying associated precursor signals. The results shed light on the mechanisms responsible for preferred propagating heatwave pathways and provide potential predictability of terrestrial heatwaves.

Terrestrial heatwaves denote extreme weather events characterized by high temperatures persisting for consecutive days or even weeks[1,2] (distinct from marine heatwaves and hereafter called "heatwaves"). These prolonged high temperatures can endanger human health[3–5], strain power systems[6], cause drought conditions, lead to crop failure[7], and increase wildfire risk[8]. The severe socioeconomic impacts have sparked great interest in the mechanisms and predictability of heatwaves[1,2]. The regions experiencing heatwaves are typically accompanied by anomalous high-pressure weather systems[1,9–12] (e.g., Supplementary Fig. S1), where the local intense heat is contributed by

adiabatic warming from subsidence, and diabatic heating from solar radiation. Operational forecasting systems can reliably issue warnings of heatwaves that last more than three days, with a lead time of two to three days[1]. Extending the lead time of heatwave forecasts requires an improved understanding of the background atmospheric circulation patterns associated with heatwaves[1,2,13,14]. On the planetary scale, atmospheric high-pressure systems are dominated by persistent anomalous atmospheric circulation patterns, such as the quasi-stationary ridge or block in the atmospheric flow[9,15], long-lived Rossby wave packets[9,10,12], or circumglobal resonant Rossby waves[16,17].

[1]Laboratory for Climate and Ocean-Atmosphere Studies, Department of Atmospheric and Oceanic Sciences, School of Physics, Peking University, Beijing, China. [2]Earth System Modelling, School of Engineering and Design, Technical University of Munich, Munich, Germany. [3]Complexity Science, Potsdam Institute for Climate Impact Research, Potsdam, Germany. [4]Center for Climate Physics, Institute for Basic Science, Busan, Republic of Korea. [5]Department of Integrated Climate System Science, Pusan National University, Busan, Republic of Korea. [6]School of Atmospheric Sciences, Sun Yat-sen University, Zhuhai, China. [7]Key Laboratory of Tropical Atmosphere-Ocean System, Ministry of Education, Zhuhai, China. [8]Southern Marine Science and Engineering Guangdong Laboratory, Zhuhai, China. [9]These authors contributed equally: Mingzhao Wang, Yu Huang. ✉e-mail: y.huang@tum.de; yuannm@mail.sysu.edu.cn; fuzt@pku.edu.cn

However, current process-based numerical models are unable to reliably predict the locations and intensities of these relevant background patterns at the desired lead times[1,18–22], which hinders both the prediction skill and our understanding of the mechanisms behind heatwaves[1,22,23]. Improved understanding of the relationship between heatwaves and these background patterns based on advanced data analysis techniques is hence of utmost importance.

Recent studies found that heatwaves in different regions overlap within a time frame of approximately one week[17,24–26], suggesting that specific anomalous atmospheric circulation patterns can establish teleconnection of hot weather events in distant locations. This phenomenon can amplify the overall socioeconomic impact of heatwaves[5,17,24,26,27], but also there might be windows of opportunity to extend the prediction of heatwaves by further understanding the spatial evolution patterns of heatwaves and the role of anomalous atmospheric circulation[1,13]. Examined on the weekly timescale, heatwaves in regions separated by 4000–5000 km can occur concurrently, which has been shown to be caused by circumglobal resonant Rossby waves[17,21]. Here, we pay attention to the daily timescale to examine the evolution of heatwaves in space and time. Taking a heatwave over central Asia as an example (Supplementary Fig. S2), the event initially affects multiple surrounding regions with hot weather (i.e., the spatial extent or size of a heatwave[28,29]), and its core gradually propagates to neighboring areas in the subsequent days. It means that heatwave events at distant regions may be linked through the heatwave's spatial extent but also by its spatial propagation. Similar phenomena can be observed in land heatwaves over Europe and North America (Supplementary Figs. S3, S4 and S5). Recent studies have confirmed that such terrestrial heatwave propagation phenomena are prevalent across global landmasses[26,30–32], but the mechanisms behind it remain mostly unknown. From the corresponding atmospheric circulation anomaly signals, it can be observed that as the heatwave propagates, the associated anomalous high-pressure systems aloft are also moving correspondingly (Supplementary Figs. S1–S5). This suggests that the propagation of heatwaves may be related to specific atmospheric circulation patterns, which deserves an in-depth investigation.

Here, we conduct a detailed analysis on the spatiotemporal evolution process of propagating heatwaves. We utilize a complex network algorithm[31,33–35] to identify the regions with the highest frequency of propagating terrestrial heatwaves in the mid-latitudes of northern hemisphere and trace their propagation pathways. We assume that such heatwave propagation phenomena are caused by more localized Rossby wave activities, as opposed to circumglobal patterns[17,21]. To verify this hypothesis, we will separately analyze the propagation of heatwaves and the movement of high-pressure atmospheric systems, then examine whether these analyses yield consistent results. Additionally, understanding the linkage with atmospheric circulation patterns can enhance the predictability of extreme weather events by providing prior knowledge from constraints on larger spatiotemporal scales, but relevant investigations for heatwaves are still lacking[1,19]. We will therefore also assess the predictability brought about by this heatwave propagation phenomenon.

## Results

### Detecting heatwaves by a complex network algorithm

To capture the spatial propagation of heatwaves across distant regions, we employ a complex network algorithm to analyze the summertime heatwave events detected in the ERA5 reanalysis (Methods). In this framework, each grid cell within the surface air temperature map is represented as a network node. For identifying the network link strength for any pair of nodes A and B, we count the number of cases that a heatwave event occurs in A, and B also experiences a heatwave event, requiring that the timing of the local temperature maximum during the heatwave in nodes A and B differ by no more than 3 days. We then perform significance tests on the link strengths,

retaining only those links that are unlikely to occur by chance[33,35,36] ("Methods"). The probability distribution of the distances of significant links exhibits two peaks, located at 1000 km and 5100 km, respectively (see Supplementary Fig. S6). The network links with typical distances of 4000-6000 km are dominated by the teleconnection effect of circumglobal resonant Rossby waves, as reported in previous studies[17,35,37,38]. Here we focus on the network links with distances of 400–2000 km, which have overall higher probabilities and could be related to more localized Rossby waves, but have been rarely analyzed in past studies. For mitigating the interference of the proximity effect[37] and spatial extent of heatwaves (with typical size of 300 km[26,29,39]) when identifying the propagation pattern of the heatwave, network links with distances shorter than 400 km are not considered during our analysis. This heatwave network is based on counting the recurrences of heatwaves in historical records using state-of-the-art data, ensuring that the subsequent network metrics capture the most frequent and recurrent patterns of heatwaves accurately.

### Propagation characteristics of heatwaves

The underlying spatial propagation patterns of heatwaves can be revealed by investigating features of the constructed heatwave network. Here, we calculate a metric called directed network divergence to analyze the characteristics of the heatwave network. Network divergence is defined as the difference between the number of outgoing links and the number of incoming links at that specific node of the network, with positive values representing the source regions of propagating heatwaves and negative values representing sinks[33,40] (see "Methods" for more details). Hence, nodes with positive divergence (source nodes) indicate that heatwaves at these nodes statistically tend to precede those in other connected nodes, while negative divergence indicates the opposite (sink nodes) (Fig. 1). On the same continent, source and sink nodes often appear in pairs, statistically indicating that in certain specific areas, such as Western Europe, Northern Asia, and North America, there may be a continuous occurrence of heatwaves from source to sink nodes. We focus on the heatwaves occurring at the network's source nodes and track their spatial evolution in the following days, finding that the heatwaves indeed propagate from the source nodes to the sink nodes (Supplementary Figs. S2–S5). Inspections of all heatwaves over the source regions identified in Fig. 1 reveal that, between 1959 and 2023, such propagating heatwaves (Supplementary Note 1) have occurred 64 times at the Northern Asia source node, 76 times at the Western Europe source node, and 83 and 79 times at the North American source nodes, respectively. This encourages us to conduct further quantitative analyses as follows.

### Preferred propagation pathways of heatwaves

Following up on the indication of source nodes in the heatwaves network, we use the "local search" method[33,36] ("Methods") to determine the main propagation directions outgoing from these source nodes, until the search stops at sink nodes. This establishes the most frequent and recurrent heatwave propagation pathways. As a result, four significant propagation pathways covering the mid-latitudes of the Northern Hemisphere are shown in Fig. 1: the Asian pathway, from the Urals to the coastal area of the Sea of Japan in East Asia; the Western European pathway (WE), from the eastern Atlantic to Central Europe (with the starting node chosen as Spain due to land-sea temperature differences); the North American Pathway 1 (NA1), extending from northern Canada to Quebec; and the North American Pathway 2 (NA2), from the west coast to the east coast of the United States. Along each pathway we select six representative nodes for subsequent analyses, with an approximate distance of 1000 km between adjacent nodes. Regarding the four detected pathways, we use rose diagrams to identify the dominant directions of the associated network links[33,36] (Supplementary Figs. S7–S10). At each node along the propagation pathway, heatwaves can spread in various directions, but the highest

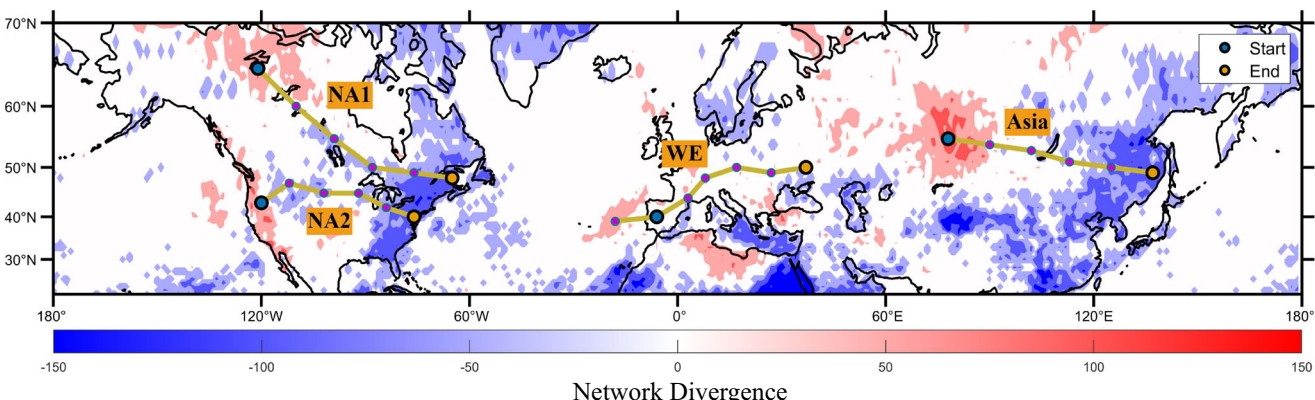

**Fig. 1 | Four preferred heatwave propagation pathways in the Northern Hemisphere from 20°N to 70°N, detected by the complex network algorithm.** Network divergence is a directed-network metric defined as the difference between the number of outgoing edges and the number of incoming edges at that specific grid cell, with positive values (red) representing the occurrence sources of events and negative values (blue) representing sinks. The four pathways (yellow) cover North America and the Eurasian continent, with starting and ending points indicated by cyan and red, respectively. The starting node of the Western Europe (WE) path is starting from the land (since we focus on land heatwaves), although strongest positive divergence over this pathway is over the ocean.

probability of propagation is distributed along the direction of the heatwave pathway, which holds for all four pathways (Supplementary Fig. S11).

## Consistency between heatwave propagation and RWPs movement

The mechanisms underlying these four significant propagation pathways are related to the atmospheric circulation background. The composite Z500 anomalies during the heatwave propagation events along the preferred pathways (Supplementary Figs. S12–S15) suggest a strong association between the propagation of these heatwaves and mid-latitude Rossby wave packets (RWPs). RWPs, as local amplitude maxima of Rossby waves, can be approximated by the centers of anomalous high-pressure systems in the upper atmosphere[10,41]. To further demonstrate the association between RWPs and the revealed heatwave propagation patterns, we thus analyze the movement trajectories of high-pressure systems using the Z500 anomaly data from ERA5. The key purpose is to compare whether the high-pressure systems' trajectories are in line with the aforesaid heatwave propagation pathways.

Taking the Asia pathway as an example, we apply the k-means clustering method to classify the movement patterns of the high-pressure system centers at the onset of heatwaves (i.e., the start node of the Asia pathway around the Ural area) into two distinct categories, and track the movement trajectories of the high-pressure system centers (Methods and Supplementary Note 2). The first category exhibits the trajectory of the moving high-pressure systems that starts at the Ural and spreads toward eastern Asia (Fig. 2), which exactly aligns with the heatwaves propagation pathway. The consistency between the heatwaves propagation pathway and the movement of high-pressure systems holds true also for the WE, NA1 and NA2 pathways (Fig. 2). In contrast, the second category remains nearly stationary near the starting node. For additional validation, we perform a similar clustering analysis on the movement of surface temperature anomaly centers and obtain two comparable categories representing moving and stationary patterns (Supplementary Fig. S16). Their trajectories also align with the high-pressure systems and the heatwave propagation pathways, further confirming the strong association between heatwaves and high-pressure systems.

The two categories of movement of high-pressure systems exhibit either propagating or standing patterns, which can further explain the spatial evolution patterns of heatwaves therein. Figure 3 shows the composite temporal evolution of surface temperature anomalies within these two categories of patterns. For the first category, the

moving high-pressure system explains the heatwave propagation (Fig. 2). High-temperature anomalies develop along the pathway within 7 days, stopping at the fifth or sixth path node (Fig. 3). Hereafter, we refer to this as the "propagation pattern" of the heatwaves. For the second category, characterised by the stationary high-pressure system, the heatwaves persist near the starting node for a longer duration (more than seven days), but its influence does not extend beyond three path nodes. This persistent heatwave could cause more severe local impacts, and we refer to this as the "standing pattern" hereafter. Similar propagating and standing patterns can be identified for the cases of the WE, NA1 and NA2 pathways (Fig. 3).

By counting the number of cases within propagating and standing patterns, respectively (Table 1), we conclude that propagating cases occupy the higher proportion of heatwaves across the four pathways. Supplementary Movies 1–4 show real-time examples of the temporal evolution of 500 hPa geopotential height and surface temperature anomalies for both propagating and standing patterns. In the propagating pattern, strong correlations are observed between the movement of high-pressure systems and the heatwaves (Supplementary Fig. S17), and the movement of the high-pressure center precedes the heatwave propagation in terms of location (see also Supplementary Note 3). A more detailed analysis will be provided in the next section.

## Enhanced Rossby wave activity flux favoring heatwave propagation

Furthermore, we investigate the energy source that sustains the propagating and standing patterns by measuring the horizontal Takaya-Nakamura Rossby wave activity flux (TN flux)[42]. For the Asia pathway, Fig. 4 presents the stream function field and TN flux vector field at local temperature maxima during the heatwave period (Day=0) and subsequently at Day = 2 and 4 under both propagation and standing scenarios. Similar results for the WE, NA1 and NA2 pathways are shown in Supplementary Figs. S18–S20. For the Asia pathway, at the moment of heatwave propagation, the TN flux indicates the flow of wave activity from the high-pressure system near the starting node toward the low-pressure system near the endpoint, representing the wave energy of RWPs. This wave activity flux causes the high-pressure system to move along the northwest-southeast direction, leading to the development of RWPs along the pathway. By Day = 2, the low-pressure system dominating the East Asian region significantly strengthens, further enhancing RWPs and ultimately intensifying the heatwaves downstream. The propagation of surface heatwaves is merely a manifestation of energy propagation in the overlaying atmosphere driven by RWPs. In contrast, for the case of the standing pattern, such energy

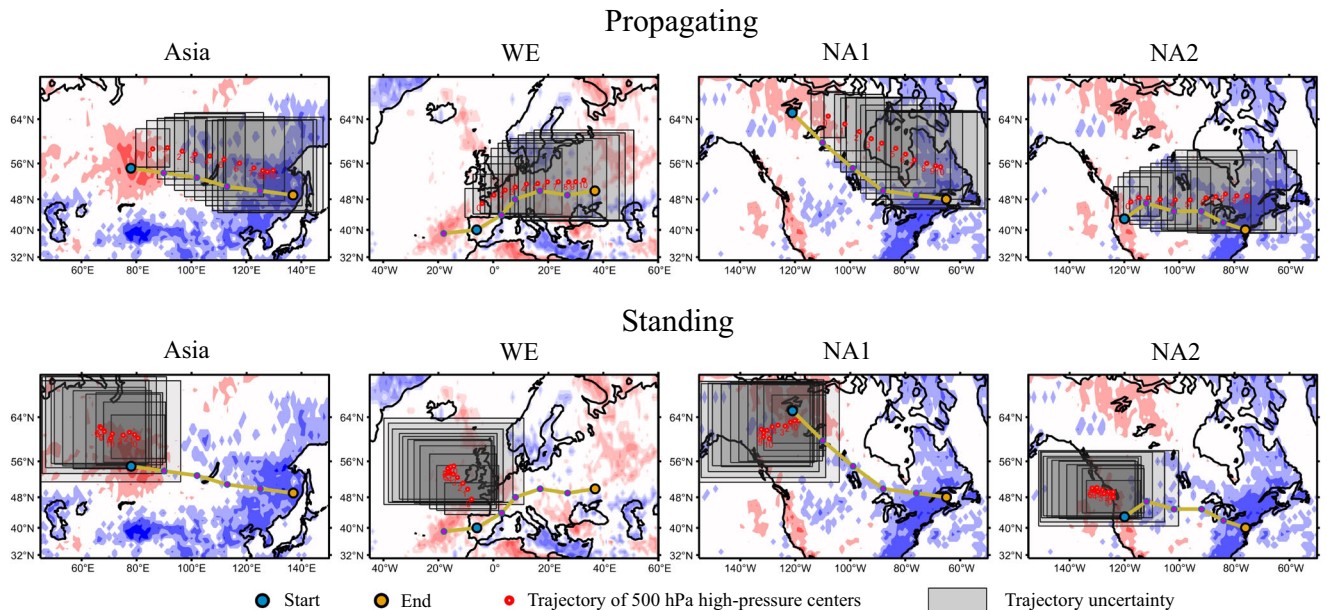

**Fig. 2 | Consistency between Rossby wave packets (RWPs) movement trajectories and heatwave propagation pathways.** For each of the four pathways, k-means clustering synthesis of the movement trajectories of 500hPa high-pressure centers 10 days after heatwaves occur at the starting node. The movement and immobility of the high-pressure center along the red trajectories align with the propagating and standing patterns of heatwaves, respectively, with shaded areas representing the uncertainty range of each point on the trajectory in the longitude/ latitude directions (Supplementary Note 2).

propagation in the overlaying atmosphere is much weaker. The TN flux exists only at the center of the high-pressure system at Day = 0, and quickly decays thereafter.

Analyses on the TN flux for the Western European and North American regions lead to similar conclusions for attributing the heatwaves propagating patterns in those regions (Supplementary Figs. S18–S20). Propagating patterns correspond to the stronger TN flux, facilitating energy propagation along the pathway and continuously triggering the generation of surface heatwaves along the pathway. Conversely, the TN flux within the standing patterns is weak, and high-pressure systems near the starting node more easily result in locally persistent, strong heatwaves.

We have also examined additional physical processes. In the propagation pattern, we observe an anomalous increase in surface shortwave radiation, with the anomaly signal propagating along four preferred pathways (Supplementary Fig. S22). The high-pressure system can cause subsiding air that warms and compresses as it descends, resulting in higher surface temperatures. In addition, the high-pressure systems bring clear skies and reduced cloud cover, allowing more shortwave radiation to reach the surface[1,2]. In contrast, for the standing pattern, the shortwave radiation anomaly remains confined to the starting node.

We also note that negative anomalies in latent heat flux propagate along both the Asia and WE pathways (Supplementary Fig. S23), indicating weakened surface evaporation. This reduction in evaporation can decrease cloud formation, allowing more shortwave radiation to reach the land surface, which further promotes the development of heatwaves[1,43]. However, for the NA1 and NA2 pathways, weak positive anomalies in latent heat flux are observed at the starting nodes, but these anomalies do not propagate along the pathways, indicating that latent heat flux does not influence heatwave propagation in these two pathways. Furthermore, heatwaves along the WE pathway show a weak connection with sea surface temperature anomalies in the eastern North Atlantic (Supplementary Fig. S24). This connection has been noted in recent studies, where the SST anomalies and land heatwaves are linked by local RWPs[44–46] (see also Supplementary Figs. S18 and S24). However, no significant

SST anomaly is observed during the analysis of other preferred propagation pathways. These analyses further confirm that intensified Rossby wave activity and RWP movement are key factors in heatwave propagation.

## Discussion

We employ a complex network algorithm to identify propagating and standing heatwave patterns in the mid-latitudes over the northern hemisphere, revealing the preferred propagation pathways of terrestrial heatwaves. Our study also unravels the association between Rossby wave packets (RWPs) and heatwave propagation. Our results are based on the ERA5 reanalysis dataset because of its state-of-the-art representation of the surface temperature and atmospheric circulations[47,48]. Future studies could benefit from additional tests using climate model simulation data and other reanalysis data for comparison.

Additionally, we note that while the frequency of heatwave occurrence along the four preferred pathways does not exceed the overall distribution across northern-hemisphere mid-latitudes (Supplementary Fig. S25a), the proportion of propagating cases is notably higher near these pathways (Supplementary Fig. S25b), and heatwave occurrences in these regions have shown increasing trends in recent decades (Supplementary Fig. S25c)[26,49]. This suggests that the four propagation pathways are located in regions with more active heatwave propagation and a rising frequency of heatwave occurrences.

Our findings concerning the four preferred propagation pathways identified here give important insights on their predictability and can provide early warning capacity for these continental-scale heatwaves and associated ecological and socioeconomic damages. For each path node along the Asia pathway (for example, but likewise for WE, NA1 and NA2 pathways), we estimate the probability of its heatwave occurrences within seven days following the onset of a heatwave event at the starting node (noted as $R_{HO}$, see Supplementary Note 4), which reflects the impact of the preferred propagation pathway on heatwave occurrence at that node[33,36]. As Fig. 5a shows, at the path nodes closest to the starting node (~1000 km), $R_{HO}$ ranges from 60% to 70% for Asia, WE, and NA1, and is close to 80% for NA2. This suggests that at the

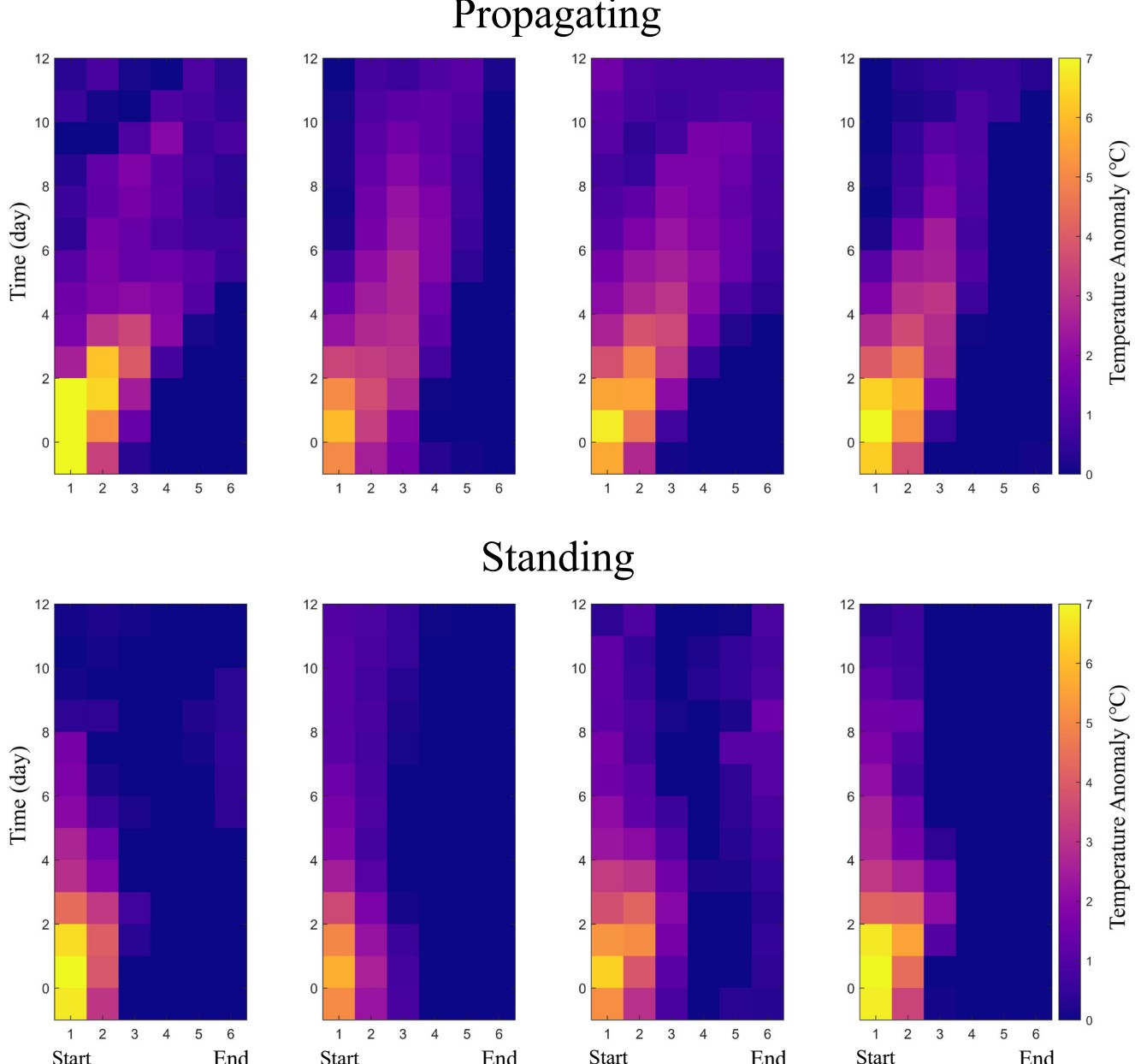

**Fig. 3 | Spatiotemporal evolutions of propagating and standing heatwaves.** Composite surface temperature anomalies as a function of time and the spatial nodes on the detected propagation pathways. The horizontal axis denotes the six representative nodes on the detected propagation pathways (see Fig. 1). Time = 0 corresponds to the timing of the local temperature maximum during the heatwave at the starting node. The standing heatwaves affect only one to two path nodes near the starting node, but their residence time at the starting node is long.

spatial scale of 1000 km, the heatwave propagation patterns can provide precursory information that could be used for early warnings of heatwave occurrences along the pathways. Figure 5b presents the case of the Asia pathway and compares the $R_{\mathrm{HO}}$ counts for three scenarios: heatwaves constrained along the pathway, heatwaves constrained along the pathway considering only propagating heatwave cases, and the $R_{\mathrm{HO}}$ count for the surrogate hypothesis test (Supplementary Note 4). For the two path nodes closest to the starting node (path nodes 1 and 2), the values of $R_{\mathrm{HO}}$ along the pathway are significant, indicating that within approximately 2000 km, which represents the characteristic spatial scale of heatwave propagation, the precursory information for the potential predictability brought by this propagation pattern is significant. This suggests that for improved predictions, heatwave forecasting should explicitly incorporate information about

upstream propagation, especially for areas with significant propagation patterns. When only propagating-heatwave cases are considered, $R_{\mathrm{HO}}$ within seven days is above 50% at path node 3. Meanwhile, heatwaves do not always propagate (see Table 1), and a considerable proportion of heatwaves belong to the standing pattern, even if they occur along propagation pathways. The proportion of propagating heatwaves across the four pathways has remained stable over recent decades (Supplementary Fig. 25d), suggesting that the predictability explained by recurrent propagation patterns is consistent over time. To further enhance the ability to predict heatwaves in practical forecasting, in the future, a deeper understanding of RWPs is needed for anticipating the onset of propagating and standing patterns earlier, and integrating the link between propagation patterns and RWPs into the machine-learning or numerical forecast models[22,23,50].

To better understand the standing heatwave patterns, we analyze the atmospheric blocking index[10,51] distributions in Supplementary Figs. S27–S30. During periods of standing heatwave patterns, atmospheric blocking can persist in a fixed area for 5–10 consecutive days, especially over North America. When a blocking system and the spatial position of heatwaves are in phase, it can prolong the duration of heatwaves, potentially leading to sustained impacts on local weather. The four regions affected by standing heatwave patterns (Fig. 2) encompass some agricultural areas, forest areas, as well as densely populated areas over the mid-latitudes of the Northern Hemisphere[52–56]. It is foreseeable that persistent standing heatwaves will have significant local impacts on health, economy, infrastructure, and ecosystems in these regions.

With regard to the potential predictability and impacts of propagating and standing patterns of heatwaves, the long-term variability and future projections of these patterns should be addressed in future research. Recent studies noticed the interannual variations in spatial extents of heatwaves[24,57]. Furthermore, the trends of propagating and standing patterns under anthropogenic global warming would be of interest, which needs in-depth analyses of CMIP6 model simulations[58]. Additionally, a more detailed understanding whether the background RWPs will exhibit interannual variability is also important to better understand.

In summary, we find that terrestrial heatwaves exhibit both propagating and standing patterns with spatial scales ranging from 400 to 2000 km. Using a complex network algorithm, four significant propagation pathways for heatwaves spanning Eurasia and North America have been identified. These patterns are guided by Rossby wave Packets. Notably, the propagation patterns provide precursor information relevant for the predictability of inter-regional heatwaves occurrences, while standing patterns are of particular relevance due to the enhanced impact of these persistent local heatwaves.

## Methods
### Data
We use 2 m temperature, 500 hPa geopotential height, 300 hPa geopotential height, horizontal wind fields, surface solar radiation, surface latent heat flux and sea surface temperature from the ERA5 reanalysis

**Table 1 | Proportion of propagating/standing heatwave events along the four preferred propagation pathways**

|             | Asia(104) | WE(140) | NA1(128) | NA2(134) |
|-------------|-----------|---------|----------|----------|
| Propagating | 61.5%     | 54.2%   | 64.8%    | 59.0%    |
| Standing    | 38.5%     | 45.8%   | 35.2%    | 41.0%    |

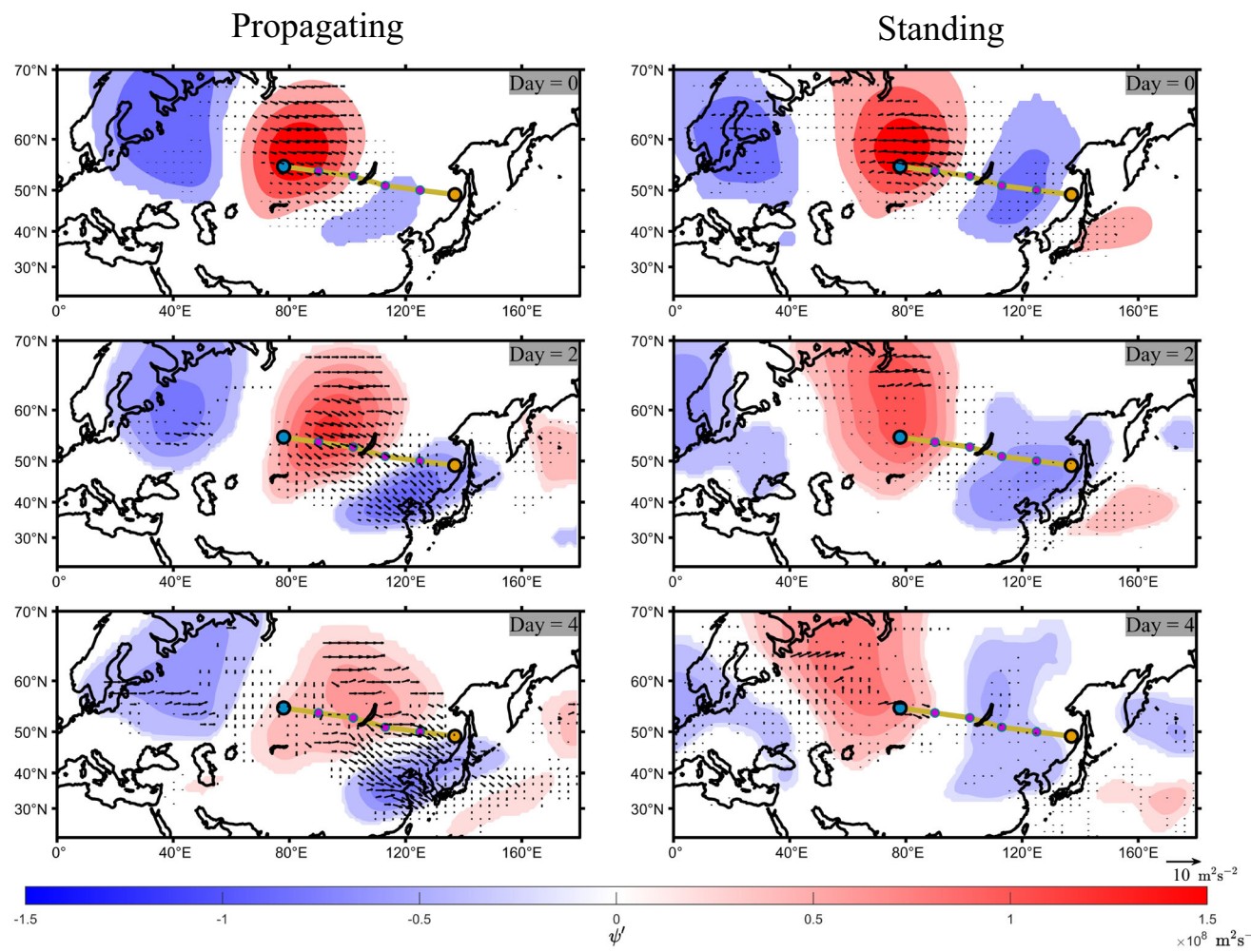

**Fig. 4 | Enhanced Rossby wave activity flux during heatwave propagations.** Colormaps present composites of the perturbed stream function field (color scale) at 300 hPa and horizontal Takaya-Nakamura Rossby wave activity flux (TN flux) for the Asian pathway after the occurrence of heatwaves at the starting node. Day = 0, 2, and 4 represent the the timing of the local temperature maximum during the heatwave at the starting node, and the subsequent 2 and 4 days, respectively. Only values significant at the 5% significance level are displayed. For the propagation heatwaves, strong TN flux is observed in the pathway region for the following 4 days, while this phenomenon is absent in the case of standing heatwaves.

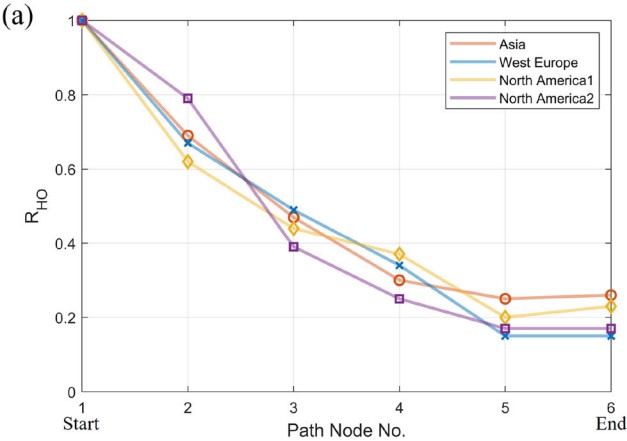

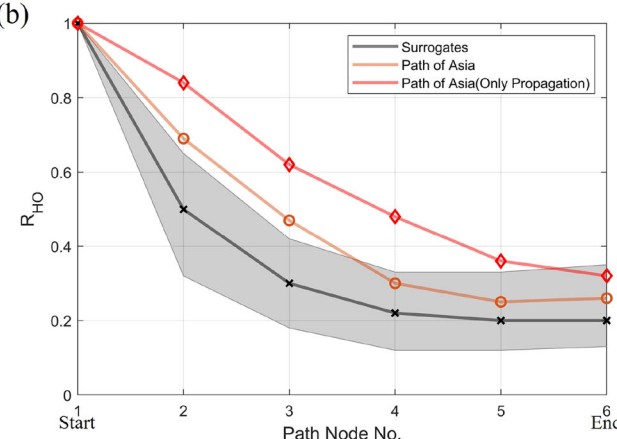

**Fig. 5 | Heatwave occurrences influenced by the preferred propagation pathways. a** Heatwave occurrence probability at different path nodes along the preferred pathways. $R_{HO}$ represents the probability of heatwave occurrence at subsequent path nodes within the 7 days following a heatwave at the starting node (Supplementary Note 4), with the probability at the starting node itself defined as 1. **b** Taking the Asian pathway as an example, heatwave occurrences are significantly influenced by the propagation effect. The orange line corresponds to the probability of heatwave occurrence within seven days as shown in (**a**), and the red line represents this probability considering only propagating-heatwave cases. Black line and shading area denote the mean and 99% confidence interval of surrogates for the significance test. The surrogates of $R_{HO}$ at each path node are calculated by counting the number of heatwaves out of the propagation pathway (Supplementary Note 4).

dataset[59] for our analysis. The spatial domain is 20°N-70°N, covering the entire longitudinal range, with a horizontal resolution of $1° \times 1°$ degree, and daily temporal resolution for the time period 1959–2023, for an extended summer season from April to September. For the time series within each spatial grid cell, we subtract its seasonal cycle and obtain the anomalies for subsequent analyses.

To define boreal summertime heatwaves, we adopt a percentile-based threshold method to identify heatwave events at each spatial grid cell[1,2]. For each cell, a heatwave event is defined as a period when the surface temperature anomaly exceeds the 95th percentile of the climatological distribution for that grid cell for more than three consecutive days, occurring between April and September[1].

### Complex network construction

We use nonlinear event synchronization (ES) measurements[40] to assess the inter-regional connection of heatwave events. The basic definition of ES remains the same as in[33,35,36], but we have made modifications in calculating network divergence. For a given pair of spatial grid cells $i$ and $j$, we calculate the normalized number of events at $i$ that can be uniquely associated with subsequent events at $j$, and vice versa: $e_i^\mu$ denotes the timing of the local temperature maximum during the $\mu$-th heatwave event at grid cell $i$, and $e_j^\nu$ correspondingly for grid cell $j$, where $\mu, \nu \in [1, l]$, $l$ denotes the total number of events at each grid cell. The two events are counted as synchronized events if their dynamical delay $d_{ij}^{\mu,\nu} : = e_i^\mu - e_j^\nu$ does not exceed a threshold $\tau$:

$$\tau = \min\left(\frac{\left\{d_{ii}^{\mu,\mu-1}, d_{ii}^{\mu,\mu+1}, d_{jj}^{\nu,\nu-1}, d_{jj}^{\nu,\nu+1}\right\}}{2}\right) \quad (1)$$

We set $S_{ij}^{\mu\nu} = 1$ if $0 < d_{ij}^{\mu,\nu} \le \tau$ and $S_{ij}^{\mu\nu} = 0$ otherwise, and calculate the normalized sum of $S_{ij}^{\mu\nu}$:

$$ES_{ij} : = \frac{\sum_{\mu\nu} S_{ij}^{\mu\nu}}{l} \quad (2)$$

Therefore, $ES_{ij}$ counts the normalized number of heatwave events traveling from $j$ to $i$, indicating the strength of the link that points from node $j$ to node $i$ in the complex network. We perform a significance test on the ES values[33,35,36]: From each node (18,000 in total), we generate

surrogate event sequences by uniformly and randomly shuffling the blocks of events. Next, we compute the ES between all nodes with randomized event sequences and, based on the histogram of these values, we determine the 0.1-confidence level for significance test on the original ES values. The ES values that are not significant are set to 0. Furthermore, to specifically examine synchronized heatwaves induced by localized Rossby waves, we restrict our analysis to a specific spatial range of network edge distances (i.e., 400–2000 km, see Supplementary Fig. S6), setting the ES values for node pairs outside this distance range to 0. The process results in a matrix that stores the modified ES values, serving as the adjacency matrix $A$ of the complex network, thereby making the network both weighted and directed.

To spatially resolve the temporal sequence of extreme events, we introduce the network divergence $\Delta D$, defined as the difference between the out-degree and in-degree for each node $i$:

$$\Delta |D_i : = D_i^{out} - D_i^{in} : = \sum_{j=1}^{N} A_{ji} - \sum_{j=1}^{N} A_{ij} \quad (3)$$

where $D_i^{out}$ and $D_i^{in}$ are the out-degree and in-degree of the node $i$. A positive $\Delta D$ indicates a source node in the network: the corresponding nodes' heatwaves are statistically followed by heatwave events at other nodes. On the contrary, a negative $\Delta D$ indicates a sink node: these nodes' heatwave events are preceded by heatwave events at other nodes.

### Local searching algorithm

After selecting the source node for heatwave propagation, we employ a local search method to identify its most representative and preferred propagation pathways[33,36]. Since heatwaves typically do not propagate strictly in a single direction but exhibit some degree of undirected divergence. This method identifies the pathway with the highest probability of heatwave propagation. Specifically, starting from the source node as the center, we conduct a search within a radius of 800-1200 km surrounding it, considering all nodes within this range. We compare the out-degree ($D_i^{out}$ in Eq.3) of each node within this circular ring around the starting node, and select the node with the highest out-degree density as the next pathway node. This process is repeated $n$ times to determine $n$ pathway nodes (Supplementary Fig. S31), where each path node serves as the center of the circular ring to identify the

subsequent path node. In our results, we set $n = 6$ to ensure that all end nodes of the pathway are located in the convergence area of the heatwave.

## Trajectory tracking algorithm

We employ a trajectory tracking method to determine the 10-day movement trajectory of the 500 hPa geopotential height the high-pressure center over a certain region following the occurrence of a heatwave (Supplementary Fig. S32). When a heatwave occurs in the selected region, we use the position of the maximum geopotential height within 1000 km of the heatwave center as the starting position of the high-pressure center (Day 0). Subsequently, using an Eulerian perspective, we track the movement of the high-pressure center for each subsequent day until the end of Day 10. Each heatwave event corresponds to a trajectory of the high-pressure center's movement.

## T-N wave-activity Flux

When investigating the intensity of the Rossby wave activity, we utilize the perturbation stream function at 300 hPa and the horizontal T-N flux[42]. The formula for the perturbation stream function $\psi'$ and the horizontal T-N Wave-Activity Flux **W** is as follows:

$$\psi' = \frac{g}{f \cdot Z'_{300hpa}} \tag{4}$$

$$\mathbf{W} = \frac{p \cos\varphi}{2|\mathbf{U}|} \begin{pmatrix} \frac{U}{a^2\cos^2\varphi}\left[\left(\frac{\partial\psi'}{\partial\lambda}\right)^2 - \psi'\frac{\partial^2\psi'}{\partial\lambda^2}\right] + \frac{V}{a^2\cos\varphi}\left[\frac{\partial\psi'}{\partial\lambda}\frac{\partial\psi'}{\partial\varphi} - \psi'\frac{\partial^2\psi'}{\partial\lambda\partial\varphi}\right] \\ \frac{U}{a^2\cos\varphi}\left[\frac{\partial\psi'}{\partial\lambda}\frac{\partial\psi'}{\partial\varphi} - \psi'\frac{\partial^2\psi'}{\partial\lambda\partial\varphi}\right] + \frac{V}{a^2}\left[\left(\frac{\partial\psi'}{\partial\varphi}\right)^2 - \psi'\frac{\partial^2\psi'}{\partial\varphi^2}\right] \end{pmatrix} \tag{5}$$

In equations (4) and (5), $U$ and $V$ represent the climatological wind fields at 300 hPa, $Z'_{300hpa}$ denotes the anomaly of geopotential height at 300 hPa, $a$ stands for the radius of the Earth, $\lambda$ and $\varphi$ represent longitude and latitude, respectively.

## Data availability

All data used in this study are openly available online. The ERA5 reanalysis dataset[59] is available at the CDS website from 1959 to 2023 (https://cds.climate.copernicus.eu/datasets). The data involved in the study have been deposited in the public repository Zenodo[60] (https://doi.org/10.5281/zenodo.15380297).

## Code availability

The code for implementing complex network algorithm, local searching, and trajectory tracking algorithms have been deposited in the public repositories Github (https://github.com/yhuangDLClimate/Propagating_heatwave and Zenodo[60] (https://doi.org/10.5281/zenodo.15380297). The analysis codes to generate the figures of this manuscript are available in Zenodo[60](https://doi.org/10.5281/zenodo.15380297).

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

## Acknowledgements

Y.H. acknowledges Alexander von Humboldt Foundation for Humboldt Research Fellowship. Z.F. and M.Z. acknowledge the National Natural Science Foundation of China (No. 42175065 and No. 42475055). C.F. was supported by the Institute for Basic Science (IBS), Republic of Korea, under IBS-R028-D1 and the National Research Fund of Korea (NRF-2022M3K3A1097082 and RS-2024-00416848). N.Y. thanks also the support from the National Natural Science Foundation of China (No. 42175068) and the support from the Guangdong Basic and Applied Basic Research Foundation (2023B1515020084). N.B. acknowledges funding by the Volkswagen Foundation. N.B. and Y.H. acknowledge the European Union's Horizon Europe research and innovation programme under grant agreement No. 101137601. This is ClimTip contribution #18.

## Author contributions

Y.H. led the study design and coordination. Y.H., M.W., N.B., C.F., Z.F. and N.Y. conceived the research. M.W. and Y.H. improved the algorithms and implemented programming. M.W. and Y.H. performed the numerical analysis. All authors interpreted and discussed the results. M.W. and Y.H. performed the visualization. Y.H. wrote the manuscript with input from all authors.

## Funding

## Competing interests

The authors declare no competing interests.
