## [Transparent Peer Review File · Nature Communications]

Evidence for preferred propagating terrestrial heatwave pathways due to Rossby wave activity

Corresponding Author: Dr Yu Huang

Version 0:

Reviewer comments:

Reviewer #1

(Remarks to the Author)

The manuscript develops complex networks based approach for understanding patterns and propagation of terrestrial heat waves and attempts to relate the latter to Rossby waves. The study highlights the potential predictability of heatwaves based on the observed propagation patterns, suggesting that these pathways could inform early warning systems. This research offers insight into the mechanics of heatwave evolution, showing that Rossby wave Pathways and the Takaya-Nakamura flux play crucial roles in the propagation process. The codes provided can be reproduced without issue. While this is an interesting work, I have a few concerns which I believe must be addressed.

1. Reproducibility is key and I would suggest that making the codes and data open access and publicly available should be a prerequisite for publication. Regarding codes, I would also request the additional codes that would allow for the reproduction of Fig. 2, Fig. 3, and Fig. 4 as part of the review process.

2. I am not too convinced that the authors can make strong statements about causal relations between land heat wave patterns and Rossby waves based on the evidence provided. Are these chance co-occurrences or are there strong mechanistic interpretations? Could data-driven causal methods be used to develop more credibility into causation assertions? In the absence of that, should the findings be appropriately caveated? I would urge the authors to consider these questions and modify the manuscript appropriately, including in the discussion sections.

3. I wonder the authors could make a stronger case by validating their findings on the relationship between heatwave propagation and Rossby wave pathways by examining a specific real-world situation, such as the South Asia heatwave that occurred this year.

4. The authors could clarify the description of the 'local Searching algorithm' (Sec. 4.3) and 'trajectory Tracking algorithm', (Sec. 4.4) along with a flowchart (this could be in a supplement if needed) to facilitate easier understanding. If these algorithms are not introduced here for the first time, consider include relevant references.

5. The authors may consider providing more information about the 'surrogate hypothesis test' in data (Sec. 4.1) or provide suitable references for readers' understanding.

(Remarks on code availability)

We have successfully reproduced all the provided codes, and they work flawlessly. We kindly request the authors to include three additional codes that would allow for the reproduction of Fig. 2, Fig. 3, and Fig. 4. Furthermore, the codes should be made open and publicly accessible with appropriate links.

Reviewer #2

(Remarks to the Author)

(Remarks on code availability)

Reviewer #3

(Remarks to the Author)

Recommendation: May be suitable for publication after major revision.

Summary:

The paper addresses an analysis of the preferred propagation pathways of boreal summer terrestrial heatwaves and its association with Rossby wave packets (RWPs). For this, the spatiotemporal evolution of heatwaves in a daily timescale is examined using a complex network algorithm (already used in other studies to identify propagation pathways), and verifying if it's aligned with the movement of high-pressure systems. Thus, the manuscript represents an effort to understand the association between RWPs and heatwaves propagation. Due to the socioeconomic impacts involved in heatwaves occurrence, the article can positively contribute to potential predictability from the propagating patterns of these extreme weather events. Some suggestions are made, which may help to further improve the work.

1. In Figure 1 is mentioned that the starting node of the Western Europe (WE) path is at the land, although the strongest positive divergence is over the ocean. However, the starting point from the land does not change the heatwave source, that is at the ocean. This could be a co-occurring terrestrial and marine heatwaves pathway? The most extreme marine heatwaves tend to occur in summer and recent investigations suggest potential interactions and common drivers between these two types of extreme events. For example, Pathmeswaran et al. (2022) found that the co-occurring terrestrial and marine heatwaves primarily occur between September and December and they are often associated with similar synoptic conditions. They also find a significant increase in the number of terrestrial heatwave days in the presence of an adjacent co-occurring marine heatwave along the coastal belt of Australia. In this sense, in Table 1 it is possible to note a smaller proportion of propagating heatwaves in WE in relation to the other pathways. Santos et al. (2024) also found co-occurring interplay between atmospheric heatwaves in southern Europe and marine heatwaves in the East Atlantic. It would be important to discuss this in the paper.

References:

Pathmeswaran C, Sen Gupta A, Perkins-Kirkpatrick SE and Hart MA (2022) Exploring Potential Links Between Co-occurring Coastal Terrestrial and Marine Heatwaves in Australia. *Front. Clim.* 4:792730. doi: 10.3389/fclim.2022.792730

Santos, R., Russo, A. & Gouveia, C.M. Co-occurrence of marine and atmospheric heatwaves with drought conditions and fire activity in the Mediterranean region. *Sci Rep* 14, 19233 (2024). <https://doi.org/10.1038/s41598-024-69691-y>

2.L.142-143: "many times over the past few decades" -> How many times during the period analyzed (1959-2023)? There are any changes over time in the heatwave's occurrence in terms of frequency? There is any propagation pathway with greater occurrence in relation to the others? It would be very interesting if the authors could briefly insert these statistics in the text.

3.L.212-213: "high consistency" -> this is more a visual verification. The authors could insert a figure with the spatial lag correlation between the positions of the high-pressure systems and the heatwaves. This could corroborate quantitatively the statement.

4.L.213-215: "the development of RWPs induces the heatwave propagation" - Does this mean that the heatwave propagation is systematically associated with a precursor RWP? Fragkoulidis et al. (2018) found that heatwaves of similar strength can be associated with highly distinctive daily evolutions of the upper-tropospheric circulation. Thus, mechanisms linking RWPs and temperature extremes are case-dependent with an important contribution of other physical mechanisms (e.g. radiative transfer, SST, subsidence, soil moisture, etc.). Thus, it is important to discuss this more deeply in the paper, as the authors did not mention the role of other mechanisms, just the movement of high-pressure systems.

Reference:

G. Fragkoulidis, V. Wirth, P. Bossmann, A. H. Fink. Linking Northern Hemisphere temperature extremes to Rossby wave packets. *Quarterly Royal of Meteorological Society*, v. 144 (711), 2018.

5.L.260-261: Any possible explanation about these differences in RHO between the pathways? Why greater for NA2?

6.L.354: "source node" – sink node? In Formula (3) and Figure 1, positive (negative) divergence represents the sources (sinks).

7.L.362: "Search within a radius of 800-1200 km" – This variable circular ring is moving from the starting point or each node

is considered as the starting point to find the next node in subsequent analyses? In Fig. S18, in legend, it is mentioned a circle of 1400-1800 km from the starting point, reaching the third node. However, if the distance between the nodes is approximately 1000 km, the most external circle would be more than 2000 km away from the original starting point. Please clarify.

8.L.375: "using a Eulerian perspective" – Why not use a Lagrangian tracking of the high-pressure centers or a quasi-Lagrangian perspective?

9. Figure 1, legend: heavewaves -> heatwaves

(Remarks on code availability)

Reviewer #4

(Remarks to the Author)
Please see the attached file.

[Editorial Note: the attachment has been appended to the very end of this file]

(Remarks on code availability)
I have briefly reviewed the code, and its format seems to be correct and functional.

Version 1:

Reviewer comments:

Reviewer #1

(Remarks to the Author)
I would like to commend the authors on meticulously addressing all my comments. I have read the full response and find their results convincing. The revised version effectively addresses three of my key comments: (a) a causality test, (b) details on their surrogate hypothesis test, and (c) validation of the findings with a specific real-world case. I would recommend acceptance.

(Remarks on code availability)
The authors have addressed all our prior comments.

Reviewer #2

(Remarks to the Author)
I co-reviewed this manuscript with one of the reviewers who provided the listed reports. This is part of the Nature Communications initiative to facilitate training in peer review and to provide appropriate recognition for Early Career Researchers who co-review manuscripts.

(Remarks on code availability)

Reviewer #3

(Remarks to the Author)
Comments, Round 2:

I really appreciated the effort made by the authors to improve the document. All my comments on the original version of the paper have been satisfactorily addressed and in my opinion the manuscript is ready for publication.

Some minor issues to be corrected:

1. Legend Fig S32 – "Central moment of the heatwave denote the the" – "Central moment of the heatwave denote the"
2. Supplementary Note 4 - P.40 – "where each of these nodes is 5at the same distance" – remove 5

(Remarks on code availability)

Reviewer #4

(Remarks to the Author)

Reviews on “Evidence for preferred propagating terrestrial heatwave pathways due to Rossby wave activity”

The authors' extensive analyses demonstrate the robustness and reproducibility of the findings, offering valuable insights into the spatial propagation mechanisms of heatwaves. Additionally, the authors have provided a well-considered response to my questions. However, I still have two remaining concerns.

1. The article concludes that the propagation paths of Rossby waves influence the trajectories of heatwaves, which may not be particularly novel. However, this finding is crucial for heatwave forecasting, given that propagation-type heatwaves account for more than 50% of cases along the four identified paths, albeit with regional variations—for instance, only around 50% for the Western European (WE) path. These regional differences could have implications for the predictability of heatwaves based on Rossby wave propagation. A key question is whether this proportion remains stable over time and what physical mechanisms govern whether a heatwave remains stationary or follows a propagation pattern in different regions. Understanding the temporal stability of propagation-type heatwaves along these paths and the underlying physical processes is essential for improving heatwave predictability.

2. The article examines the relationship between Rossby wave propagation and heatwave forecasting. Since temperature-based forecasting (traditional numerical weather forecasting) already provides a certain lead time (7-10 days) for predicting heatwaves, another key question is, compared to temperature-based methods, what are the advantages of using Rossby wave propagation for improving heatwave prediction?

Typo errors:

1. Abstract “fix stations” should be “fixed stations”
2. Line 54 three days two days in advance

(Remarks on code availability)

Version 2:

Reviewer comments:

Reviewer #4

(Remarks to the Author)

Thank you to the authors for carefully addressing my previous comments. I believe the paper is now suitable for publication.

(Remarks on code availability)

RESPONSE TO REVIEWER COMMENTS

Please find our point-by-point responses to the comments raised by the referee in blue font below.

Reviewer 1

The manuscript develops complex networks based approach for understanding patterns and propagation of terrestrial heat waves and attempts to relate the latter to Rossby waves. The study highlights the potential predictability of heatwaves based on the observed propagation patterns, suggesting that these pathways could inform early warning systems. This research offers insight into the mechanics of heatwave evolution, showing that Rossby wave Pathways and the Takaya-Nakamura flux play crucial roles in the propagation process. The codes provided can be reproduced without issue. While this is an interesting work, I have a few concerns which I believe must be addressed.

Thank you very much for the thorough review of our paper and for providing us with very helpful suggestions. We also appreciate the recognition of the interest and value of our work. We have carefully taken your comments into account and have made revisions to address each of the points you highlighted. Please find detailed-point-by-point responses to your concerns, as well as references to how we revised our manuscript, in the following.

1. Reproducibility is key and I would suggest that making the codes and data open access and publicly available should be a prerequisite for publication. Regarding codes, I would also request the additional codes that would allow for the reproduction of Fig. 2, Fig. 3, and Fig. 4 as part of the review process.

Thank you for your suggestion. We are now sharing the code and raw data used to compute and generate Figs. 1, 2, 3, 4 and 5 in the Code Ocean capsule. We have updated the Code Availability section to indicate that our codes will be published on Github/Zenodo for public access following the publication of the manuscript.

2. I am not too convinced that the authors can make strong statements about causal relations between land heat wave patterns and Rossby waves based on the evidence provided. Are these chance co-occurrences or are there strong mechanistic interpretations? Could data-driven causal methods be used to develop more credibility into causation assertions? In the absence of that, should the findings be appropriately caveated? I would urge the authors to consider these questions and modify the manuscript appropriately, including in the discussion sections.

Thank you, we agree. Existing studies have interpreted that localized Rossby Wave Packets (RWPs) have significant causal impact on heatwaves, playing an important role in the underlying physical mechanisms (Refs. 1-4): during summertime, RWPs are accompanied with local atmospheric high-pressure systems or atmospheric blocking highs, and such high-pressure systems influence heatwaves by causing subsiding air that warms and compresses as it descends, resulting in higher surface temperatures. Additionally, the high-pressure systems bring clear skies and reduced cloud cover, allowing more shortwave radiation to reach the surface. Existing statistical studies (Refs. 4-5) have also found a close connection between RWPs and heatwaves in both the Northern and Southern Hemispheres.

In our revised manuscript, following your suggestion, we have applied the Granger causality test (Refs. 6) to investigate the causal relationship between heatwaves (HWs) and Rossby wave packets (RWPs) along the propagation pathway. Taking the starting node as an example, we first compare the time series of the surface air temperature anomaly (SATA), geopotential height anomaly at 500 hPa (z500) and the absolute amplitude of TN flux for Rossby wave activity (WAF amplitude) during heatwaves (Fig. S21). Both local maxima of the WAF amplitude and z500 precede that of the SATA, and this holds for all four preferred pathways. Similar conclusions can be drawn from the analysis on time-lagged correlations between z500 and SATA at different path nodes (Fig. S17), where z500 precedes SATA by 1 day during the period of propagating heatwaves. This suggests that the RWP precedes HW during the period of propagating heatwaves (Refs. 7-8). We further conduct the aforesaid Granger causality test on the paired time series of z500 and SATA, as well as WAF amplitude and SATA. The test shows that the Granger causal influence from z500 to SATA is significant, while the Granger causal influence from SATA to z500 is not significant. Similarly, it is also found that WAF amplitude unidirectionally Granger-causes SATA. Similar causal inference results are found during our analysis on other path nodes.

Adapted from Supplementary Fig. S21: Comparison of composite time series during propagating heatwaves for the geopotential height anomaly at 500 hPa (z500), the absolute amplitude of TN flux for Rossby wave activity (WAF amplitude), and surface air temperature anomaly (SATa) at the starting nodes of the Asia, WE, NA1, and NA2 heatwave pathways, respectively. Time = 0 corresponds to the timing of the local temperature maximum during the heatwave at the starting node. To facilitate comparison, the time series of each variable were normalized to have a mean of 0 and a standard deviation of 1.

Adapted from Supplementary Fig. S17: Time-lagged Pearson correlation between temperature and z500 anomalies at individual path nodes during periods of propagating heatwaves, with results presented for the Asia, WE, NA1, and NA2 pathways, respectively. Positive time lag indicates that the z500 anomaly is lagged relative to the temperature anomaly. Only cases of propagating heatwaves are included in this analysis. The z500 anomaly is highly correlated (with Pearson correlation higher than 0.8) with the temperature anomaly at time lag = 0, and this holds for all path nodes. Specifically, at time lag = -1, where the z500 anomaly precedes the temperature anomaly by 1 day, the first three path nodes exhibit a maximum correlation (higher than 0.94) between the temperature and z500 anomalies, indicating that the z500 anomaly leads the temperature anomaly.

In the revised manuscript, we have included the Granger causality analyses of Rossby waves and heatwaves in Supplementary Note 3. Additionally, we have incorporated an analysis of other physical processes, such as surface shortwave radiation and surface latent heat flux (lines 243-265), and found that, compared to these other processes, RWPs are the key factor driving heatwave propagation patterns.

3. I wonder the authors could make a stronger case by validating their findings on the relationship between heatwave propagation and Rossby wave pathways by examining a specific real-world situation, such as the South Asia heatwave that occurred this year.

Thank you for your suggestion. We added a case for the propagating heatwave occurring in Southern and Eastern Asia in the year of 2022 (Supplementary Fig. S1). On June 10th, 2022, a heatwave was detected in the Urals and Central Asia, with high temperatures gradually spreading to Southern and Eastern Asia over the following 10 days. During the development, the location of the positive center of 500 hPa geopotential height anomalies keeps overlapping with that of the high temperature center. The spatial domain of our study covers the mid-latitudes of the Northern Hemisphere, ranging from 20°N to 70°N, and does not extend to the subtropical region of South Asia like India and Pakistan. Additionally, recent studies (Refs. 1, 9 and 10) suggest that the spatial spread of heatwaves in the subtropical region of South Asia is related to the late-monsoon effect and the strong local northerly wind anomalies. In the revised manuscript, we have added the case of the propagating heatwave occurring at the south and middle of East Asia in the year of 2022 in Supplementary Fig. S1.

Adapted from Supplementary Fig. S1: On June 10, 2022, the distribution of surface temperature and 500 hPa geopotential height anomalies (contours) across Asia. Days 0 to 10 represent the evolution of surface temperature anomalies from the onset of the heatwave to 10 days thereafter, illustrating a complete process of heatwave propagation from the Urals to the south and middle of East Asia.

4. The authors could clarify the description of the ‘local Searching algorithm’ (Sec. 4.3) and ‘trajectory Tracking algorithm’, (Sec. 4.4) along with a flowchart (this could be in a supplement if needed) to facilitate easier understanding. If these algorithms are not introduced here for the first time, consider include relevant references.

Thank you for your suggestion. In supplementary Figs. S31 and S32 of the revised manuscript, we have implemented the flowcharts for ‘Local Searching algorithm’ and ‘Trajectory Tracking algorithm’, respectively. The method of ‘Local Searching algorithm’ was initially developed by Boers et al. 2014 (Ref. 11) and Li et al., 2024 (Ref. 12), and we have added the corresponding references in Methods.

5. The authors may consider providing more information about the ‘surrogate hypothesis test’ in data (Sec. 4.1) or provide suitable references for readers’ understanding.

Thank you for your suggestion. In the revised manuscript, we have added a more detailed explanation for the surrogate hypothesis test in Supplementary Note 4, including the precise definition of R_{HO} and the proper revision on Supplementary Fig. S26, as below:

*“Supplementary Note 4: Calculation and **surrogate hypothesis test** for R_{HO}*

We define an index R_{HO} to assess the impact of each preferred propagation pathway on heatwave occurrences downstream. Taking the Asia pathway as an example, R_{HO} represents the probability of heatwave occurrences at the k -th path node within 7 days following a heatwave at the starting node, where $k \in [1,6]$ represents the order of the six representative nodes along the pathway (Fig. 1).

$$R_{HO}^k = \frac{N_{1|k}}{N_k} \quad (5)$$

where N_k is the total number of heatwave events at the k -th path node, and $N_{1|k}$ denotes the number of cases that a heatwave occurs at the k -th path node within 7 days following a heatwave at the starting node. Therefore, for each path node, R_{HO}^k is used to estimate the impact of the preferred propagation pathway on heatwave occurrence at that node (Fig. 5a).

*To verify the significance of the preferred propagation pathway's impact on heatwave occurrences at its downstream nodes, **we use surrogates to compute R_{HO} and test the null hypothesis**. Specifically, when a heatwave occurs at the starting node of the Asia pathway, we estimate the probability of a heatwave occurring at different nodes within the following 7 days (i.e., R_{HO}), where each of these nodes is at the same distance from the starting node.*

Thus, these nodes are distributed in a circle centered on the starting node (Supplementary Fig. S26): one of these nodes lies along the Asia pathway, with the probability of heatwave occurrence denoted as $R_{HO}(\text{Path})$; while the remaining nodes, which lie outside the Asia pathway, serve as surrogates, with their probability of heatwave occurrence denoted as $R_{HO}(\text{Non-path})$. Hence, the null hypothesis states that $R_{HO}(\text{Path})$ will not differ significantly from $R_{HO}(\text{Non-path})$.

We calculate $R_{HO}(\text{Non-path})$ values for the surrogates, and obtain the mean, 99th percentile and 1st percentile of them. The results show that $R_{HO}(\text{Path})$ is beyond the 99% confidence

interval for the surrogates (as in Fig. 5), confirming the significant impact of preferred propagation pathway on the heatwave occurrence at its downstream nodes. Similar significance tests using surrogates were also performed for the West Europe, North America 1, and North America 2 pathways (not shown here), where the impacts of these pathways were found to be significant as well.”

Adapted from Supplementary Fig. S26. Illustration of surrogate generation for the R_{HO} significance test. Taking the 3rd path node of the Asia pathway as an example, we consider all grid cells at a similar distance from the starting node, comparable to the distance between the starting node and the 3rd path node (around 1600 km). These grid cells, represented by the red dots within black rings with radii of 1400 km and 1800 km, are not part of the Asia pathway but serve as surrogates for the 3rd path node in subsequent significance tests. The brown arrow indicates that the R_{HO} calculation is performed on the 3rd path node of the Asia pathway, and the result is denoted as $R_{HO}(\text{Path})$. The gray arrow, instead, indicates that the R_{HO} calculation is performed on the surrogates, denoted as $R_{HO}(\text{Non-path})$. See Supplementary Note 4 for the details of calculating R_{HO} .

Reviewer #1 (Remarks on code availability):

We have successfully reproduced all the provided codes, and they work flawlessly. We kindly request the authors to include three additional codes that would allow for the reproduction of Fig. 2, Fig. 3, and Fig. 4. Furthermore, the codes should be made open and publicly accessible with appropriate links.

Thank you. We are now sharing all code and raw data used to compute and generate Figs. 1, 2, 3, 4 and 5 in the Code Ocean capsule. We have updated the Code Availability section to

indicate that our codes will be published on Github/Zenodo for public access following the publication of the manuscript.

Reference

Ref. 1: Domeisen, D.I., Eltahir, E.A., Fischer, E.M., Knutti, R., Perkins-Kirkpatrick, S.E., Schar, C., Seneviratne, S.I., Weisheimer, A., Wernli, H.: Prediction and projection of heatwaves. *Nature Reviews Earth & Environment* 4(1), 36–50410 (2023)

Ref. 2: Barriopedro, D., Garcia-Herrera, R., Ordonez, C., Miralles, D., Salcedo-Sanz, S.: Heat waves: Physical understanding and scientific challenges. *Reviews of Geophysics* 61(2), 2022–000780 (2023)

Ref. 3: Ma, Q., Franzke, C.L.: The role of transient eddies and diabatic heating in the maintenance of european heat waves: a nonlinear quasi-stationary wave perspective. *Climate Dynamics* 56(9-10), 3003–3004 (2021)

Ref. 4: Risbey, J.S., O’Kane, T.J., Monselesan, D.P., Franzke, C.L., Horenko, I.: On the dynamics of austral heat waves. *Journal of Geophysical Research: Atmospheres* 123(1), 38–57 (2018)

Ref. 5: Fragkoulidis, G., Wirth, V., Bossmann, P., Fink, A.: Linking northern hemisphere temperature extremes to Rossby wave packets. *Quarterly Journal of the Royal Meteorological Society* 144(711), 553–566 (2018)

Ref. 6: Granger, C.W.: Investigating causal relations by econometric models and cross-spectral methods. *Econometrica: journal of the Econometric Society*, 424–438 (1969)

Ref. 7: McGraw, M.C., Barnes, E.A.: Memory matters: A case for granger causality in climate variability studies. *Journal of climate* 31(8), 3289–3300 (2018)

Ref. 8: Huang, Y., Franzke, C.L., Yuan, N., Fu, Z.: Systematic identification of causal relations in high-dimensional chaotic systems: application to stratosphere-troposphere coupling. *Climate Dynamics* 55, 2469–2481 (2020)

Ref. 9: Röthlisberger, M., Sprenger, M., Flaounas, E., Beyerle, U., Wernli, H.: The substructure of extremely hot summers in the Northern Hemisphere. *Weather and Climate Dynamics*, 1(1), 45-62 (2020)

Ref. 10: Dar, J.A., Apurv, T.: Spatiotemporal characteristics and physical drivers of heatwaves in india. *Geophysical Research Letters* 51(15), 2024–109785 (2024)

Ref. 11: Boers, N., Bookhagen, B., Barbosa, H.M., Marwan, N., Kurths, J., Marengo, J.A.: Prediction of extreme floods in the eastern central andes based on a complex networks approach. *Nature Communications* 5, 5199 (2014)

Ref. 12: Li, K., Huang, Y., Liu, K., Wang, M., Cai, F., Zhang, J., Boers, N.: Key propagation pathways of extreme precipitation events revealed by climate networks. *npj Climate and Atmospheric Science* 7(1), 165 (2024)

Reviewer 2

Thank you very much for the very thoughtful co-review of our manuscript and for providing us with very helpful suggestions. We have carefully revised our manuscript according to your comments. Please find our detailed-point-by-point responses to your comments and suggestions, as well as references to how we revised our manuscript accordingly.

Reviewer 3

The paper addresses an analysis of the preferred propagation pathways of boreal summer terrestrial heatwaves and its association with Rossby wave packets (RWPs). For this, the spatiotemporal evolution of heatwaves in a daily timescale is examined using a complex network algorithm (already used in other studies to identify propagation pathways), and verifying if it's aligned with the movement of high-pressure systems. Thus, the manuscript represents an effort to understand the association between RWPs and heatwaves propagation. Due to the socioeconomic impacts involved in heatwaves occurrence, the article can positively contribute to potential predictability from the propagating patterns of these extreme weather events. Some suggestions are made, which may help to further improve the work.

Thank you very much for the accurate summary of our study and the recognition of the interest and value of our work. Your suggestions have been very valuable in improving our work. Please find point-by-point responses to your comments, as well as references to how we revised our manuscript, in the following.

1. In Figure 1 is mentioned that the starting node of the Western Europe (WE) path is at the land, although the strongest positive divergence is over the ocean. However, the starting point from the land does not change the heatwave source, that is at the ocean. This could be a co-occurring terrestrial and marine heatwaves pathway? The most extreme marine heatwaves tend to occur in summer and recent investigations suggest potential interactions and common drivers between these two types of extreme events. For example, Pathmeswaran et al. (2022) found that the co-occurring terrestrial and marine heatwaves primarily occur between September and December and they are often associated with similar synoptic conditions. They also find a significant increase in the number of terrestrial heatwave days in the presence of an adjacent co-occurring marine heatwave along the coastal belt of Australia.

In this sense, in Table 1 it is possible to note a smaller proportion of propagating heatwaves in WE in relation to the other pathways. Santos et al. (2024) also found co-occurring interplay between atmospheric heatwaves in southern Europe and marine heatwaves in the East Atlantic. It would be important to discuss this in the paper.

References:

Pathmeswaran C, Sen Gupta A, Perkins-Kirkpatrick SE and Hart MA (2022) Exploring Potential Links Between Co-occurring Coastal Terrestrial and Marine Heatwaves in Australia. *Front. Clim.* 4:792730. doi: 10.3389/fclim.2022.792730

Santos, R., Russo, A. & Gouveia, C.M. Co-occurrence of marine and atmospheric heatwaves with drought conditions and fire activity in the Mediterranean region. *Sci Rep* 14, 19233 (2024). <https://doi.org/10.1038/s41598-024-69691-y>

Thank you for your suggestion. We presented the composited sea surface temperature (SST) anomalies corresponding to heatwaves along the Western Europe pathway and found the co-existing SST anomalies in the North Atlantic Ocean and heatwaves in Western Europe (Supplementary Fig. S24), although the composited SST anomalies are relatively weak (around 0.5°C). In the propagating pattern, the SST anomalies also show a slight eastward movement from Day = -4 to Day = 6 (Supplementary Fig. S24a). Additionally, the patterns of SST anomalies differ between the standing and propagating heatwave patterns. A recent study (Ref. 1) also found that European heatwaves could be linked to North Atlantic SST anomalies, with high-pressure systems and local Rossby waves playing key roles in bridging this connection. Our focus is on the propagating pathway of the heatwaves, and the SST anomalies observed here need further investigation in future work.

Adapted from Supplementary Fig. S24: Composite sea surface temperature anomaly during (a) the propagating heatwaves and (b) the standing heatwaves on the Western Europe pathway. Only values significant at the 5% significance level are displayed.

In lines 258-265 of the revised manuscript, we have properly included the discussion about the North Atlantic SST anomalies, as below:

“Furthermore, heatwaves along the WE pathway show a weak connection with sea surface temperature anomalies in the eastern North Atlantic (Supplementary Fig. S24). This connection has been noted in recent studies, where SST anomalies and land heatwaves are linked by local RWPs [42–44] (see also Supplementary Figs. S18 and S24). However, no significant SST anomaly is observed during the analysis of other preferred propagation pathways. These analyses further confirm that intensified Rossby wave activity and RWP movement are key factors in heatwave propagation.”

2. L.142-143: “many times over the past few decades” -> How many times during the period analyzed (1959-2023)? There are any changes over time in the heatwave’s occurrence in terms of frequency? There is any propagation pathway with greater occurrence in relation to the others? It would be very interesting if the authors could briefly insert these statistics in the text.

Thank you for your suggestion. Between 1959 and 2023, the propagating heatwaves have occurred 64 times at the Northern Asia source node, 76 times at the Western Europe source node, and 83 and 79 times at the North American source nodes, respectively. We found significantly increasing trends of the heatwave occurrence frequency in the Northern Hemisphere and also the regions near preferred pathways (Supplementary Fig. S25c). We also inspected the frequency of heatwave occurrences in the mid-latitudes of the Northern Hemisphere, and the proportion of the cases of propagating heatwaves. The frequency of heatwave occurrence along these four pathways does not exceed the overall distribution across northern-hemisphere mid-latitudes (Supplementary Fig. S25a), but the proportion of propagating cases is notably higher near these pathways (Supplementary Fig. S25b).

In the revised manuscript, we have revised the descriptions in lines 142-146, and incorporated the analyses on frequency of heatwave occurrences in lines 275-282, and Supplementary Fig. 25, as follows:

“Inspections of all heatwaves over the source regions identified in Fig. 1 reveal that, between 1959 and 2023, such propagating heatwaves (Supplementary Note 1) have occurred 64 times at the Northern Asia source node, 76 times at the Western Europe source node, and 83 and 79

times at the North American source nodes, respectively. This encourages us to conduct further quantitative analyses as follows” (lines 142-146), and “Additionally, we note that while the frequency of heatwave occurrence along the four preferred pathways does not exceed the overall distribution across northern-hemisphere mid-latitudes (Supplementary Fig. S25a), the proportion of propagating cases is notably higher near these pathways (Supplementary Fig. S25b), and heatwave occurrences in these regions have shown increasing trends in recent decades (Supplementary Fig. S25c) [24, 47]. This suggests that the four propagation pathways are located in regions with more active heatwave propagation and a rising frequency of heatwave occurrences.” (lines 275-282).

Adapted from Supplementary Fig. S25: Statistics of heatwave occurrences (1959-2023) in the Northern Hemisphere from 20°N to 70°N, and comparison with the preferred propagation pathways of heatwaves revealed in our study (yellow lines). (a) Frequency of heatwave occurrence; (b) proportion of the cases of propagating heatwaves (Supplementary Note 1); and (c) trend of heatwave occurrence frequency.

3. L.212-213: “high consistency” -> this is more a visual verification. The authors could insert a figure with the spatial lag correlation between the positions of the high-pressure systems and the heatwaves. This could corroborate quantitatively the statement.

Thank you for your suggestion. To visualize the consistency between the movement of heatwaves and high-pressure systems along the preferred pathway, we examine the time-lagged Pearson correlation between temperature and Z500 anomalies at individual path nodes during propagating heatwave periods, as shown in Fig. S17. Taking the Asia pathway as an example, the Z500 anomaly is highly correlated (Pearson correlation > 0.8) with the temperature anomaly at time lag = 0, and this holds for all path nodes. Specifically, at time lag = -1, where the Z500 anomaly precedes the temperature anomaly by 1 day, the first three path nodes exhibit a maximum correlation (Pearson correlation > 0.94) between the temperature and Z500 anomalies, indicating that the Z500 anomaly leads the temperature anomaly. Similar conclusions can be drawn from examination on the WE, NA1 and NA2 heatwave pathways. In the revised manuscript, we have included the time lagged Pearson correlations at individual path nodes into Supplementary Fig. S17.

Adapted from Supplementary Fig. S17: Time-lagged Pearson correlation between temperature and Z500 anomalies at individual path nodes during periods of propagating heatwaves, with results presented for the Asia, WE, NA1, and NA2 heatwave pathways, respectively. Positive time lag indicates that the Z500 anomaly is lagged relative to the temperature anomaly. Only cases of propagating heatwaves are included in this analysis. The Z500 anomaly is highly correlated (with Pearson correlation higher than 0.8) with the temperature anomaly at time lag = 0, and this holds for

all path nodes. Specifically, at time lag = -1, where the Z500 anomaly precedes the temperature anomaly by 1 day, the first three path nodes exhibit a maximum correlation (higher than 0.94) between the temperature and Z500 anomalies, indicating that the Z500 anomaly leads the temperature anomaly.

4. L.213-215: “the development of RWPs induces the heatwave propagation” - Does this mean that the heatwave propagation is systematically associated with a precursor RWP? Fragkoulidis et al. (2018) found that heatwaves of similar strength can be associated with highly distinctive daily evolutions of the upper-tropospheric circulation. Thus, mechanisms linking RWPs and temperature extremes are case-dependent with an important contribution of other physical mechanisms (e.g. radiative transfer, SST, subsidence, soil moisture, etc.). Thus, it is important to discuss this more deeply in the paper, as the authors did not mention the role of other mechanisms, just the movement of high-pressure systems.

Reference:

G. Fragkoulidis, V. Wirth, P. Bossmann, A. H. Fink. Linking Northern Hemisphere temperature extremes to Rossby wave packets. *Quarterly Royal of Meteorological Society*, v. 144 (711), 2018.

Thank you for your helpful suggestion. In our manuscript, we examine the Takaya-Nakamura Rossby wave activity flux in the upper troposphere (at the 300 hPa pressure level), which distinguishes between the propagation and standing patterns of heatwaves. Additionally, we conducted analyses on additional physical processes, as follows:

(1) In the propagation pattern, we observe an anomalous increase in surface shortwave radiation, with the anomaly signal propagating along four preferred pathways (Supplementary Fig. S22). The high-pressure systems can cause subsiding air that warms and compresses as it descends, resulting in higher surface temperatures. Additionally, the high-pressure systems bring clear skies and reduced cloud cover, allowing more shortwave radiation to reach the surface. In contrast, for the standing pattern, the shortwave radiation anomaly remains confined to the starting node.

Adapted from Supplementary Fig. S22: Spatiotemporal evolutions of surface solar radiation anomaly during propagating and standing heatwaves. Composite surface solar radiation anomaly as a function of time and the spatial nodes on the detected propagation pathways. Horizontal axis denotes the six representative nodes on the four detected propagation pathways (see Fig. 1). Time = 0 corresponds to the timing of the local temperature maximum during the heatwave at the starting node.

(2) We also note that negative anomalies in latent heat flux propagate along both the Asia and WE pathways (Supplementary Fig. S23), indicating weakened surface evaporation. This reduction in evaporation can decrease cloud formation, allowing more shortwave radiation to reach the land surface, which further promotes heatwave development. However, for the NA1 and NA2 pathways, weak positive anomalies in latent heat flux are observed at the starting nodes, but these anomalies do not propagate along the pathways, indicating that latent heat flux does not influence heatwave propagation in these two pathways.

Adapted from Supplementary Fig. S23: Spatiotemporal evolutions of surface latent heat flux anomalies during propagating and standing heatwaves. Composite surface latent heat flux anomaly as a function of time and the spatial nodes on the four detected propagation pathways. The horizontal axis denotes the six representative nodes on the detected propagation pathways (see Fig. 1). Time = 0 corresponds to the timing of the local temperature maximum during the heatwave at the starting node.

(3) Furthermore, heatwaves along the WE pathway show a weak connection with sea surface temperature anomalies in the eastern North Atlantic (Supplementary Fig. S24). This connection has been noted in recent studies (Refs. 5-6), where SST anomalies and land heatwaves are linked by local RWPs (see also Supplementary Figs. S18 and S24). However, no significant SST anomaly is observed during the analysis of other preferred propagation pathways.

These analyses further confirm that intensified Rossby wave activity and RWP movement are key factors in heatwave propagation of the four preferred pathways.

Adapted from Supplementary Fig. S24: Composite sea surface temperature anomaly during (a) the propagating heatwaves and (b) the standing heatwaves on the Western Europe pathway. Only values significant at the 5% significance level are displayed.

In the revised manuscript, we have incorporated the analyses on the additional physical processes in lines 243–265, as follows:

“We have also examined additional physical processes. In the propagation pattern, we observe an anomalous increase in surface shortwave radiation, with the anomaly signal propagating along four preferred pathways (Supplementary Fig. S22). The high-pressure system can cause subsiding air that warms and compresses as it descends, resulting in higher surface temperatures. In addition, the high-pressure systems bring clear skies and reduced cloud cover, allowing more shortwave radiation to reach the surface [1, 2]. In contrast, for the standing pattern, the shortwave radiation anomaly remains confined to the starting node. We also note that negative anomalies in latent heat flux propagate along both the Asia and WE pathways (Supplementary Fig. S23), indicating weakened surface evaporation. This reduction in evaporation can decrease cloud formation, allowing more shortwave radiation to reach the land surface, which further promotes the development of heatwaves [1, 41]. However, for the NA1 and NA2 pathways, weak positive anomalies in latent heat flux are observed at the starting nodes, but these anomalies do not propagate along the pathways, indicating that latent heat flux does not influence heatwave propagation in these two pathways. Furthermore, heatwaves along the WE pathway show a weak connection with sea surface temperature anomalies in the eastern North Atlantic (Supplementary Fig. S24). This connection has been noted in recent studies, where SST anomalies and land heatwaves are linked by local RWPs [42–44] (see also Supplementary Figs. S18 and S24). However, no

significant SST anomaly is observed during the analysis of other preferred propagation pathways. These analyses further confirm that intensified Rossby wave activity and RWP movement are key factors in heatwave propagation.”

5. L260-261: Any possible explanation about these differences in RHO between the pathways? Why greater for NA2?

R_{HO} represents the probability of heatwave occurrences at a path node within seven days following the onset of a heatwave event at the starting node. The R_{HO} for the second path node in the NA2 pathway (around 80%) is a bit higher than in other pathways (around 70%). However, from the third to the sixth path nodes in the NA2 pathway, the R_{HO} decays more rapidly, falling below the values observed in the other pathways. This may be due to the slower propagation of heatwaves in the NA2 pathway (Fig. 3), allowing the heatwaves to persist longer around the second path node compared to the other pathways. This prolonged presence can facilitate the increase of the R_{HO} values at the second path node. Additionally, the R_{HO} values shown in Fig. 5a include all heatwave cases, which also account for heatwaves from the standing pattern category. In this category, heatwaves at the second path node of the NA2 pathway can persist for a longer duration than those in other pathways (bottom right of Fig. 3). As a result, more heatwave events at the second path node of the NA2 pathway are included in the R_{HO} than in other pathways. Heatwave occurrences can also be influenced by the local land surface conditions in different continents. The comparison across different continents is complex and warrants further investigation in future studies.

Adapted from Fig. 3: Spatiotemporal evolutions of propagating and standing heatwaves. Composite surface temperature anomalies as a function of time and the spatial nodes on the detected propagation pathways. The horizontal axis denotes the six representative nodes on the detected propagation pathways (see Fig. 1). Time = 0 corresponds to the timing of the local temperature maximum during the heatwave at the starting node.

6. L.354: “source node” – sink node? In Formula (3) and Figure 1, positive (negative) divergence represents the sources (sinks).

Sorry, this was a typo, and we have modified it in line 381 of the revised manuscript. Thank you for the reminder.

7. L.362: “Search within a radius of 800-1200 km” – This variable circular ring is moving from the starting point or each node is considered as the starting point to find the next node in subsequent analyses? In Fig. S18, in legend, it is mentioned a circle of 1400-1800 km from the starting point, reaching the third node. However, if the distance between the nodes is

approximately 1000 km, the most external circle would be more than 2000 km away from the original starting point. Please clarify.

Thank you for your suggestion. In the method for identifying the preferred propagation pathway, each path node is treated as the center of the circular ring, then to identify the next node in subsequent analyses. This method was initially developed by Boers et al. 2014 (Ref. 2) and Li et al., 2024 (Ref. 3), and we have properly added the corresponding references in Methods.

For clarity, in the revised manuscript, we have modified the method introduction in lines 394-398 and added a flowchart in Supplementary Fig. S31, illustrating this method for identifying the preferred propagation pathway, as shown below:

“This process is repeated n times to determine n pathway nodes (Supplementary Fig. S31), where each path node serves as the center of the circular ring to identify the subsequent path node. In our results, we set $n = 6$ to ensure that all end nodes of the pathway are located in the convergence area of the heatwave.”

Adapted from Supplementary Fig. S31: Flowchart diagram illustrating the local searching algorithm for identifying preferred propagation pathway of heatwaves in the complex network.

Fig. S18 in the original manuscript was intended to explain the surrogate generation and significance test of R_{HO} , rather than the method for identifying the preferred propagation pathway. Fig. S18 used the third path node as an example, illustrating how to perform the significance test for the impact of the source node on heatwave occurrence at the third path node. The distance between the source node and the third path node is approximately 1600

km. Therefore, the black rings with radii of 1400 km and 1800 km are used to identify surrogate nodes at similar distances from the source node to the third path node.

In the revised manuscript, Fig. S18 in the previous manuscript has been moved to Fig. S26. For clarification, we have modified the figure caption and added Supplementary Note 4 to explain this figure in more detail, as below:

“Supplementary Note 4: Calculation and surrogate hypothesis test for R_{HO}

We define an index R_{HO} to assess the impact of each preferred propagation pathway on heatwave occurrences downstream. Taking the Asia pathway as an example, R_{HO} represents the probability of heatwave occurrence at the k -th path node within 7 days following a heatwave at the starting node, where $k \in [1,6]$ represents the order of the six representative nodes along the pathway (Fig. 1).

$$R_{HO}^k = \frac{N_{1|k}}{N_k} \quad (5)$$

N_k is the total number of heatwave events at the k -th path node, and $N_{1|k}$ denotes the number of cases that a heatwave occurs at the k -th path node within 7 days following a heatwave at the starting node. Therefore, for each path node, R_{HO}^k is used to estimate the impact of the preferred propagation pathway on heatwave occurrence at that node (Fig. 5a).

To verify the significance of the preferred propagation pathway's impact on heatwave occurrences at its downstream nodes, we use surrogates to compute R_{HO} and test the null hypothesis. Specifically, when a heatwave occurs at the starting node of the Asia pathway, we estimate the probability of a heatwave occurring at different nodes within the following 7 days (i.e., R_{HO}), where each of these nodes is at the same distance from the starting node.

Thus, these nodes are distributed in a circle centered on the starting node (Supplementary Fig. S26): one of these nodes lies along the Asia pathway, with the probability of heatwave occurrence denoted as $R_{HO}(\text{Path})$; while the remaining nodes, which lie outside the Asia pathway, serve as surrogates, with their probability of heatwave occurrence denoted as $R_{HO}(\text{Non-path})$. Hence the null hypothesis states that $R_{HO}(\text{Path})$ will not differ significantly from $R_{HO}(\text{Non-path})$.

We calculate $R_{HO}(\text{Non-path})$ values for the surrogates, and obtain the mean, 99th percentile and 1st percentile of them. Result shows $R_{HO}(\text{Path})$ is beyond the 99% confidence interval for

the surrogates (as in Fig. 5), confirming the significant impact of preferred propagation pathway on the heatwave occurrence at its downstream nodes. Similar significance tests using surrogates were also performed for the West Europe, North America 1, and North America 2 pathways (not shown here), where the impacts of these pathways were found to be significant as well.”

Adapted from Supplementary Fig. S26. Illustration of surrogate generation for the R_{HO} significance test. Taking the 3rd path node of the Asia pathway as an example, we consider all grid cells at a similar distance from the starting node, comparable to the distance between the starting node and the 3rd path node (around 1600 km). These grid cells, represented by the red dots within black rings with radii of 1400 km and 1800 km, are not part of the Asia pathway but serve as surrogates for the 3rd path node in subsequent significance tests. The brown arrow indicates that the R_{HO} calculation is performed on the 3rd path node of the Asia pathway, and the result is denoted as $R_{HO}(\text{Path})$. Whereas gray arrow indicates that the R_{HO} calculation is performed on the surrogates, denoted as $R_{HO}(\text{Non-path})$. See Supplementary Note 4 for the details of calculating R_{HO} .

8. L.375: “using a Eulerian perspective” – Why not use a Lagrangian tracking of the high-pressure centers or a quasi-Lagrangian perspective?

Thank you. The Lagrangian perspective could be an alternative approach, but it typically demands high-resolution data and could be computationally intensive (e.g., Ref. 4). On the other hand, here our complex network algorithm is based on the Eulerian perspective. We believe that using the Eulerian perspective here better maintains the consistency and

simplicity of the methodology. A comparison of our results to an approach based on Lagrangian tracking would be a very interesting subject of future work.

9. Figure 1, legend: heavewaves -> heatwaves

Thank you. We have modified it in the revised manuscript.

Reference

Ref. 1: Duchez, A., Frajka-Williams, E., Josey, S.A., Evans, D.G., Grist, J.P., Marsh, R., McCarthy, G.D., Sinha, B., Berry, D.I., Hirschi, J.J.: Drivers of exceptionally cold north atlantic ocean temperatures and their link to the 2015 european heatwave. *Environmental Research Letters* 11(7), 074004 (2016)

Ref. 2: Boers, N., Bookhagen, B., Barbosa, H.M., Marwan, N., Kurths, J., Marengo, J.A.: Prediction of extreme floods in the eastern central andes based on a complex networks approach. *Nature Communications* 5, 5199 (2014)

Ref. 3: Li, K., Huang, Y., Liu, K., Wang, M., Cai, F., Zhang, J., Boers, N.: Key propagation pathways of extreme precipitation events revealed by climate networks. *npj Climate and Atmospheric Science* 7(1), 165 (2024)

Ref. 4: Papritz L, Röthlisberger M. A novel temperature anomaly source diagnostic: method and application to the 2021 heatwave in the Pacific Northwest. *Geophysical Research Letters*. 50, 23: e2023GL105641 (2023)

Ref. 5: Santos, R., Russo, A., Gouveia, C.M.: Co-occurrence of marine and atmospheric heatwaves with drought conditions and fire activity in the mediterranean region. *Scientific Reports* 14(1), 19233 (2024)

Ref. 6: Duchez, A., Frajka-Williams, E., Josey, S.A., Evans, D.G., Grist, J.P., Marsh, R., McCarthy, G.D., Sinha, B., Berry, D.I., Hirschi, J.J.: Drivers of exceptionally cold north atlantic ocean temperatures and their link to the 2015 european heatwave. *Environmental Research Letters* 11(7), 074004 (2016)

Reviewer 4

This work investigates the mechanisms underlying the spatial propagation of the terrestrial heatwaves and find out four preferred propagation pathways of terrestrial heatwaves in the northern hemisphere. And each preferred pathways are consistent with the movement of Rossby wave trains. The detected propagation pathways are found to provide prior knowledge for occurrences of downstream heatwaves and thus can be used for identifying associated precursor signals. I find the results interesting and innovative, as they provide a compelling explanation for the occurrence of simultaneous extremely high temperatures across the globe within a short period. This insight sheds light on the underlying mechanisms that drive these widespread heatwave events. However, the method and expressions are a little hard to follow. Please see the detailed comments below.

Thank you very much for taking the time to review our paper and for offering valuable suggestions. We greatly appreciate the recognition of the interest and value of our work. In response, we have carefully revised the manuscript and made improvements to the Methods section. Below, we provide a detailed point-by-point response to your comments and suggestions, along with references to the corresponding revisions in the manuscript.

Overall, the method parts are very complex and confusing,

1. Definition of Heatwaves: The authors state in the manuscript, “A heatwave with the spatial extent (around 300 km [26]) could be detected by multiple surrounding regions, and this is taken into account when analyzing heatwaves by the used complex network algorithm.” However, it remains unclear how many hot regions a heatwave comprises or what the minimum spatial extent should be for an event to qualify as a heatwave. Could the authors clarify these points to ensure a consistent and precise definition of heatwaves in their analysis?

Thank you for this comment. We apologize, this could have been much clearer. In our study and methodology, we identify a heatwave event using the traditional definition (Ref. 1): “at each spatial grid cell, if its surface temperature anomaly exceeds the 95th percentile of the climatological distribution for that grid cell for more than three consecutive days and occurs between April and September, it is considered a heatwave event”. The spatial size of heatwave was not used for identifying heatwave events at each spatial grid cell in our study.

We note that heatwave events at different spatial grid cells may be linked through the heatwave's spatial extent but also by its spatial propagation, but our study merely focuses on the heatwave's spatial propagation. Therefore, in the subsequent complex network algorithm, it is important to note that only the average or maximum spatial extents are considered, as they help mitigate the interference of the heatwave's spatial extent when identifying the heatwave's propagation pattern. Earlier studies (Ref. 2 and Ref. 3) suggested that the minimum, maximum and average spatial extent of a heatwave are 151000 km^2 (219 km in radius), 400000 km^2 (356 km in radius) and 300000 km^2 (309 km in radius), respectively, with the typical size being approximately 300 km in radius. Therefore, we set a cut-off distance of 400 km, analyzing only the complex network's links with distances greater than 400 km. The minimum spatial extent is not involved during the analysis. We have revised the sentences in lines 347-351, 78-80 and 119-122 of the manuscript, as follows:

“Definition of boreal summertime heatwaves: Here we adopt the percentile-based threshold method to identify heatwave events at each spatial grid cell [1, 2]. For each spatial grid cell, a heatwave event is defined as a period when the surface temperature anomaly exceeds the 95th percentile of the climatological distribution for that grid cell for more than three consecutive days, occurring between April and September [1]” (lines 347-351),

“It means that heatwave events at distant regions may be linked through the heatwave's spatial extent but also by its spatial propagation” (lines 78-80),

and *“For mitigating the interference of the proximity effect [32] and spatial extent of heatwaves (with typical size of 300 km [26, 27, 34]) when identifying the heatwave's propagation pattern, network links with distances shorter than 400 km are not considered during our analysis”* (lines 119-122).

2. The calculation of ES measurements: I am confused about the definition of μ, v, e and d . They are not clearly explained, which make the method hard to be understood. More specific, what exactly do the event time series e_i/e_j represent? The average temperature or maximum temperature or the start date/end date for a heatwave event?

Thank you for raising this point. For a given pair of spatial grid cells i and j , e_i^μ denotes the timing of the local temperature maximum during the μ -th heatwave event in grid cell i , and e_j^v correspondingly for grid cell j , where $\mu, v \in [1, l]$, l denotes the total number of heatwave events at each grid cell. $d_{i,j}^{\mu,v}$ is equal to $e_i^\mu - e_j^v$, and denotes the time delay between these two events. We have improved the descriptions in lines of 355-361 of the revised manuscript:
“For a given pair of spatial grid cells i and j , we calculate the normalized number of events at i that can be uniquely associated with subsequent events at j , and vice versa: e_i^μ denotes the timing of the local temperature maximum during the μ -th heatwave event at grid cell i , and e_j^v correspondingly for grid cell j , where $\mu, v \in [1, l]$, l denotes the total number of events at each grid cell. The two events are counted as synchronized events if their dynamical delay $d_{i,j}^{\mu,v} := e_i^\mu - e_j^v$ does not exceed a threshold τ ”

3. Does the matrix A consist of modified ES values ?

Yes, the modified ES values are stored in matrix A . We have improved the sentences in lines 374-376 of the revised manuscript:

“This process results in a matrix that stores the modified ES values, serving as the adjacency matrix A of the complex network, thereby making the network both weighted and directed.”

4. Local Searching algorithm: how is the out-degree obtained?

Thank you. The out-degree is calculated from the network's adjacent matrix A , as in Eq.3. In the revised manuscript, we have supplemented the descriptions in lines 380 and 392-394:

“where D_i^{out} and D_i^{in} are the out-degree and in-degree of the node i ” (line 380),

and *“We compare the out-degree (D_i^{out} in Eq.3) of each node within this circular ring around the starting node”*(line 392-394).

5. Also, I am confused about the expression of heatwaves in “region A or B” and “node A and B”.

Thank you. The terms "regions A and B" are intended to have the same meaning as "nodes A and B". We have modified this expression as "nodes A and B" throughout the revised manuscript.

The result descriptions:

6. As stated in Line 142 “Inspections of all heatwaves over the source regions identified from Fig. 1 suggests that such propagating heatwaves have occurred many times over the past few decades.” Could you be more specific, or provide a global map for the frequency of heatwave occurrences.

Thank you. Between 1959 and 2023, propagating heatwaves have occurred 64 times at the Northern Asia source node, 76 times at the Western Europe source node, and 83 and 79 times at the North American source nodes, respectively. Supplementary Fig. S25 illustrates the frequency of heatwave occurrences in the mid-latitudes of the Northern Hemisphere, and the proportion of the cases of propagating heatwaves. The frequency of heatwave occurrence along these four pathways does not exceed the overall distribution across northern-hemisphere mid-latitudes (Supplementary Fig. S25a), but the proportion of propagating cases is notably higher near these pathways (Supplementary Fig. S25b).

Adapted from Supplementary Fig. S25: Statistics of heatwave occurrences (1959-2023) in the Northern Hemisphere from 20°N to 70°N, and comparison with the preferred propagation pathways of heatwaves revealed in our study (yellow lines). (a) Frequency of heatwave occurrence, and (b) proportion of the cases of propagating heatwaves (Supplementary Note 1).

In the revised manuscript, we have revised the descriptions in lines 142-146, and incorporated an analysis of the frequency of heatwave occurrences in lines 275-282, and Supplementary Fig. S25:

“Inspections of all heatwaves over the source regions identified in Fig. 1 reveal that, between 1959 and 2023, such propagating heatwaves (Supplementary Note 1) have occurred 64 times at the Northern Asia source node, 76 times at the Western Europe source node, and 83 and 79 times at the North American source nodes, respectively. This encourages us to conduct further quantitative analyses as follows” (lines 142-146),

and *“Additionally, we note that while the frequency of heatwave occurrence along the four preferred pathways does not exceed the overall distribution across northern-hemisphere mid-latitudes (Supplementary Fig. S25a), the proportion of propagating cases is notably higher near these pathways (Supplementary Fig. S25b), and heatwave occurrences in these regions have shown increasing trends in recent decades (Supplementary Fig. S25c) [24, 47]. This suggests that the four propagation pathways are located in regions with more active heatwave propagation and a rising frequency of heatwave occurrences”* (lines 275-282).

7. Figure 1 is interesting. It gives the four typical heatwave propagation pathways across the whole globe. The pathway over Asia indicates that heatwaves over the eastern coast regions of China generally originate from Ural. I am curious about where the heatwaves in southern China originate. Do they originate from the same source as the eastern part of China? Additionally, it would be helpful to clarify how much these four typical propagation pathways contribute to the overall heatwave pathways worldwide.

Thank you for your suggestion. Some of the heatwaves in the southern and middle part of China can be related to the Ural: Taking the case of June 2022 as an example (Supplementary Fig. S1), for a heatwave in the Ural, the subsequent spatial movement of high temperature anomalies involves a broad spatial range over eastern and southern China. This suggests that heatwaves can spread in various directions around the Asia pathway. But most of the

directions (61.5%) are toward the southeast and align with the extension of the Asia pathway, as shown by the percentages in the rose diagram of movement directions for heatwaves at the Asia pathway (Supplementary Fig. S11). As designed by the algorithm, the spatial extension of the preferred Asia pathway is along the direction with the highest probability of movement direction, illustrating the predominant direction and representative pathway for most heatwave movements. For the preferred pathways in Western Europe and North America we created similar rose diagrams to present the percentages of movement directions of heatwaves in their respective regions, as shown in Supplementary Fig. S11.

Adapted from Supplementary Fig. S1 On June 10, 2022, the distribution of surface temperature and 500 hPa geopotential height anomalies (contours) across Asia. Days 0 to 10 represent the evolution of surface temperature anomalies from the onset of the heatwave to 10 days thereafter, illustrating a complete process of heatwave propagation from the Ural to the south and middle of East Asia.

Adapted from Supplementary Fig. S11 Rose diagrams of movement directions for the heatwaves at starting nodes of the four preferred pathways (Asia, WE, NA1 and NA2 pathways), respectively. The dots connected with yellow lines denote the spatial routes of the four preferred pathways.

In the revised manuscript, we added Fig. S1 to present the case of the heatwave propagating to the southern and middle part of China. We also added Supplementary Fig. S11 to illustrate the percentages of movement directions of heatwaves, compared with the extension directions of preferred pathways in Asia, Western Europe and North America, respectively.

8. How to obtain the centers of the 500-hPa pressure system and their uncertainty range in Figure 2?

Taking the Asia pathway as an example, our procedure is:

- (1) For each heatwave event, we first identify the geographic location with the maximum of the 500-hPa anomaly around the starting node (on Day = 0, i.e., the central moment (the day of local temperature maximum) of the heatwave event), designating this as the center of the high-pressure system on Day = 0.
- (2) For the subsequent days, we identify the centers of the high-pressure system from Day = 1 to Day = 10, allowing us to track the movement of the high-pressure system throughout the heatwave event.

(3) This process yields the ensemble movement trajectories for all heatwave events, and we apply k-means clustering to categorize them into propagating and standing patterns.

(4) For each day's set of trajectory points, we calculate the average geographic location as the center. The uncertainty for each point is determined as twice the standard deviation of its latitude and longitude, as shown in Fig. 2.

In the revised manuscript, we have supplemented this method description into Supplementary Note 2:

“Supplementary Note 2: Identifying trajectory and uncertainty of the high-pressure system movement

Figure 2 presents the trajectory and uncertainty of the high-pressure system. Taking the Asia pathway as an example, our procedure is as follows:

(1) For each heatwave event, we first identify the geographic location with the maximum of the 500-hPa anomaly around the starting node (on Day = 0, i.e., the central moment (the day of local temperature maximum) of the heatwave event), designating this as the center of the high-pressure system on Day = 0.

(2) For the subsequent days, we identify the centers of the high-pressure system from Day = 1 to Day = 10, allowing us to track the movement of the high-pressure system throughout the heatwave event.

(3) This process yields the ensemble movement trajectories for all heatwave events, and we apply k-means clustering to categorize them into propagating and standing patterns.

(4) For each day's set of trajectory points, we calculate the average geographic location as the center. The uncertainty for each point is determined as twice the standard deviation of its latitude and longitude, as shown in Fig. 2.”

9. Line 186-187 The stationary heatwaves are firstly mentioned here but without a detail description. Please clarify how to distinguish the propagating and stationary heatwaves?

Thank you for this suggestion. The stationary heatwaves were identified through clustering analysis on the movement trajectories of the high-pressure system's center:

(1) When we performed unsupervised k-means clustering on the movement trajectories of the high-pressure system's center, we obtained two categories: one exhibits long-traveling behavior, and another exhibits nearly stationary behavior (as shown in Fig. 2). (2) To

double-check, we applied the same clustering analysis but to the surface air temperature's highest value center, and obtained long-traveling and stationary categories as well (as shown in Supplementary Fig. S16). (3) Next, we composited the surface air temperature anomalies corresponding to these two categories of the high-pressure system's movement, and found that the composition of the "stationary-behavior" category shows the pattern of stationary heatwaves (as shown in Fig. 3), and the composition of the "long-traveling-behavior" category shows the pattern of propagating heatwaves that is in line with the analysis result by using complex network.

In the revised manuscript, we have improved the descriptions in lines 179-193:

"Taking the Asia pathway as an example, we apply the k-means clustering method to classify the movement patterns of the high-pressure system centers at the onset of heatwaves (i.e., the start node of the Asia pathway around the Ural area) into two distinct categories, and track the movement trajectories of the high-pressure system centers (Methods and Supplementary Note 2). The first category exhibits the trajectory of the moving high-pressure systems that starts at the Ural and spreads towards eastern Asia (Fig. 2), which exactly aligns with the heatwaves propagation pathway. The consistency between the heatwaves propagation pathway and the movement of high-pressure systems holds true also for the WE, NA1 and NA2 pathways (Fig. 2). In contrast, the second category remains nearly stationary near the starting node. For additional validation, we perform a similar clustering analysis on the movement of surface temperature anomaly centers and obtain two comparable categories representing moving and stationary patterns (Supplementary Fig. S16). Their movement trajectories also align with the high-pressure systems and the heatwave propagation pathways, further confirming the strong association between heatwaves and high-pressure systems."

10. How is RHO calculated? Is it a distance or a ratio? So confused.

We apologize, this could have been much clearer. Indeed, R_{HO} is a ratio, not a distance, and it quantifies the impact of each preferred propagation pathway on downstream heatwave occurrences. We adopted this quantifier based on the work of Boers et al. (2014) and Li et al. (2024) (Refs. 4 and 5). In the revised manuscript, we have added a more detailed explanation,

including the precise definition and mathematical formulation of R_{HO} in Supplementary Note 4:

“Supplementary Note 4: Calculation and significant test for R_{HO}

We define an index R_{HO} to assess the impact of each preferred propagation pathway on heatwave occurrences downstream. Taking the Asia pathway as an example, R_{HO} represents the probability of heatwave occurrence at the k -th path node within 7 days following a heatwave at the starting node, where $k \in [1,6]$ represents the order of the six representative nodes along the pathway (Fig. 1).

$$R_{HO}^k = \frac{N_{1|k}}{N_k} \quad (5)$$

where N_k is the total number of heatwave events at the k -th path node, and $N_{1|k}$ denotes the number of cases that a heatwave occurs at the k -th path node within 7 days following a heatwave at the starting node. Therefore, for each path node, R_{HO}^k is used to estimate the impact of the preferred propagation pathway on heatwave occurrence at that node (Fig. 5a).”

11. How can we recognize the duration of blocking system from Fig. S19?

We presented the spatiotemporal distribution of blocking indices for the standing heatwave patterns in the four pathways, respectively (Supplementary Figs. S27-S30). Taking the standing heatwave patterns in the Asia pathway as an example (Supplementary Fig. S27), the negative blocking index values around Ural denote the pattern of a blocking system, and this pattern can persist from Day = -4 to Day = 2, hence the duration time of this blocking system is approximately 6 days. For the WE, NA1 and NA2 pathways, a similar analysis is conducted.

In the revised manuscript, we have added the spatiotemporal distribution of the blocking index for the standing heatwave patterns in the four pathways, respectively, in Supplementary Figs. S27-S30.

Adapted from Supplementary Fig. S27: Spatiotemporal distribution of GHGN blocking index (Supplementary Note 4) for the standing heatwave patterns in the Asia pathway. Day=0, 2, and 4 respectively represent the central moment (the date of local maxima) of heatwave occurrence at the starting node and the subsequent 2 and 4 days. Day= -4 and -2 denote the 2 and 4 days before the central moment of the heatwave. A more negative GHGN value indicates a stronger blocking.

Reviewer #4 (Remarks on code availability):

I have briefly reviewed the code, and its format seems to be correct and functional.

Thank you. We are now sharing all code and raw data used to compute and generate Figs. 1, 2, 3, 4 and 5 in the Code Ocean capsule. We have updated the Code Availability section to indicate that our codes will be published on Github/Zenodo for public access following the publication of the manuscript.

Reference

Ref. 1: Domeisen D.I., Eltahir E.A., Fischer E.M., Knutti R., Perkins-Kirkpatrick S.E., Schar C., Seneviratne S.I., Weisheimer A., Wernli H.: Prediction and projection of heatwaves. *Nature Reviews Earth & Environment* 4(1), 36–50414 (2023).

Ref. 2: Keellings D., Bunting E., Engström J.: Spatiotemporal changes in the size and shape of heat waves over North America. *Climatic Change* 147, 165–178 (2018).

Ref. 3: Luo M., Wu S., Lau G.N.-C., Pei T., Liu Z., Wang X., Ning G., Chan T.O., Yang Y., Zhang W.: Anthropogenic forcing has increased the risk of longer-traveling and slower-moving large contiguous heatwaves. *Science Advances* 10(13), 1598 (2024).

Ref. 4: Boers, N., Bookhagen, B., Barbosa, H.M., Marwan, N., Kurths, J., Marengo, J.A.: Prediction of extreme floods in the eastern central andes based on a complex networks approach. *Nature Communications* 5, 5199 (2014)

Ref. 5: Li, K., Huang, Y., Liu, K., Wang, M., Cai, F., Zhang, J., Boers, N.: Key propagation pathways of extreme precipitation events revealed by climate networks. *npj Climate and Atmospheric Science* 7(1), 165 (2024)

RESPONSE TO REVIEWER COMMENTS

Please find our point-by-point responses to the comments raised by the referee in blue font below.

Reviewer 1

I would like to commend the authors on meticulously addressing all my comments. I have read the full response and find their results convincing. The revised version effectively addresses three of my key comments: (a) a causality test, (b) details on their surrogate hypothesis test, and (c) validation of the findings with a specific real-world case. I would recommend acceptance.

Thank you for your valuable feedback and recognition. We are pleased to hear that you are satisfied with the revised version and appreciate your recommendation for acceptance.

Reviewer #1 (Remarks on code availability):

The authors have addressed all our prior comments.

Thank you.

Reviewer 2

Thank you for the thoughtful co-review of our manuscript and for providing us with helpful suggestions. Please find our detailed-point-by-point responses to your comments and suggestions, as well as references to how we revised our manuscript accordingly.

Reviewer 3

Comments, Round 2:

I really appreciated the effort made by the authors to improve the document. All my comments on the original version of the paper have been satisfactorily addressed and in my opinion the manuscript is ready for publication.

Thank you for your valuable feedback and recognition. We are pleased to hear that you are satisfied with the revised version and appreciate your recommendation for acceptance.

Some minor issues to be corrected:

1. Legend Fig S32 – “Central moment od the heatwave denote the the” – “Central moment of the heatwave denote the”

Thanks. We have corrected this typo in the revised manuscript.

2. Supplementary Note 4 - P.40 – “where each of these nodes is 5at the same distance” – remove 5

Thanks. We have corrected this typo.

Reviewer 4

Reviews on “Evidence for preferred propagating terrestrial heatwave pathways due to Rossby wave activity”

The authors' extensive analyses demonstrate the robustness and reproducibility of the findings, offering valuable insights into the spatial propagation mechanisms of heatwaves. Additionally, the authors have provided a well-considered response to my questions. However, I still have two remaining concerns.

Thank you for your valuable feedback and recognition. We have carefully considered your comments and suggestions. Please find below our detailed point-by-point responses, along with references to the corresponding revisions made in the manuscript.

1. The article concludes that the propagation paths of Rossby waves influence the trajectories of heatwaves, which may not be particularly novel. However, this finding is crucial for heatwave forecasting, given that propagation-type heatwaves account for more than 50% of cases along the four identified paths, albeit with regional variations—for instance, only around 50% for the Western European (WE) path. These regional differences could have implications for the predictability of heatwaves based on Rossby wave propagation. A key question is whether this proportion remains stable over time and what physical mechanisms govern whether a heatwave remains stationary or follows a propagation pattern in different regions. Understanding the temporal stability of propagation-type heatwaves along these paths and the underlying physical processes is essential for improving heatwave predictability.

Thank you for this suggestion. (1) We analyzed the temporal trend of propagating heatwave proportions at each grid cell in the Northern Hemisphere, from 20°N to 70°N, covering the period from 1959 to 2023. Most of the trends were statistically insignificant, indicating that the proportion of propagating heatwaves has remained stable over time. This temporal stability of the proportion holds for most grid cells and across the four preferred pathways (Supplementary Fig. S25d).

(2) The identified heatwave propagation pathways can help to improve the heatwaves' predictability downstream of the pathway. We agree that understanding the onset of propagation patterns is crucial for further enhancing heatwave predictability over the pathway's start point and also the entire pathway. Our findings show that the propagating and

stationary patterns of heatwaves are closely linked to the movement of high-pressure systems associated with Rossby wave packets (Figs. 2 and 4). Additionally, we observed that land soil moisture exerts the feedback effect on the propagation of heatwaves (Supplementary Fig. S23). To thoroughly understand and predict the onset and development of the propagation and standing patterns, a comprehensive numerical-simulation experiment would be needed. However, current physical-processes-based climate models are unable to correctly predict the location, strength and movement direction of the Rossby wave trains associated with heatwaves at desired lead times (see Refs. 1-4). On the other hand, as previous studies suggested (e.g., Refs. 1, 4, 5 and 6), topography, surface friction and soil moisture play a critical role in determining the ability of numerical climate models to characterize the stability, frequency, and transition phases of Rossby wave packets, thereby influencing their predictability (e.g., Refs. 3, 4 and 6). The performance to reproduce the link between Rossby wave packets and topography in state-of-the-art numerical weather prediction models and climate models also remains an open question (e.g., Refs. 1, 4 and 5). Therefore, it is challenging to anticipate the onset of the propagating and standing patterns earlier. A comprehensive research addressing this question would be a very interesting subject of future work.

In the revised manuscript, we have added a figure illustrating the trend analysis of the proportion of the propagating heatwaves, as shown in Supplementary Fig. 25d. We also have incorporated the relevant discussion in lines 60-63, and lines 309-316:

Lines: 60-63: “However, current process-based numerical models are unable to reliably predict the locations and strengths of these relevant background patterns at the desired lead times [1, 18–22], which hinders both the prediction skill and our understanding of the mechanisms behind heatwaves[1, 22, 23].”

Lines 309-316: “The proportion of propagating heatwaves across the four pathways has remained stable over recent decades (Supplementary Fig. 25d), suggesting that the predictability explained by recurrent propagation patterns is consistent over time. To further enhance the ability to predict heatwaves in practical forecasting, in the future, a deeper understanding of RWPs is needed for anticipating the onset of propagating and standing patterns earlier, and integrating the link between propagation patterns and RWPs into the machine-learning or numerical forecast models [22, 23, 50].”

Adapted from Supplementary Figure 25d: Statistics of heatwave occurrences (1959-2023) in the Northern Hemisphere from 20°N to 70°N, and comparison with the preferred propagation pathways of heatwaves revealed in our study (yellow lines). (a) Frequency of heatwave occurrences; (b) proportion of the cases of propagating heatwaves (Supplementary Note 1); (c) trend of the frequency of heatwave occurrences; and (d) trend of the proportion of the propagating heatwaves. Cross marks indicate trend values that are significant at the 5% significance level.

2. The article examines the relationship between Rossby wave propagation and heatwave forecasting. Since temperature-based forecasting (traditional numerical weather forecasting) already provides a certain lead time (7-10 days) for predicting heatwaves, another key question is, compared to temperature-based methods, what are the advantages of using Rossby wave propagation for improving heatwave prediction?

Thank you for raising this important point. As recent studies have suggested, numerical weather forecasting systems can reliably issue the warning of heatwave, with a lead time of 2

to 3 days (Ref. 1); however, as the forecast lead time is further extended, the accuracy of heatwave predictions significantly decreases (Refs. 1, 3, and 7). This decline is primarily due to the inadequate representation of the evolution of Rossby waves and the relationship between Rossby wave patterns and heatwaves in numerical weather forecasting and climate models (Refs. 1, 3, 4, and 7). Specifically, it was found that the current numerical forecast models are unable to predict the location, strength, and movement direction of local Rossby wave packets associated with the heatwaves at the desired lead times (Refs. 1, 2 and 3). Our study reveals the association between propagating heatwaves pathways and the recurrent Rossby waves patterns based on real-world cases. Integrating or reproducing this linkage in the state-of-the-art numerical models or the machine-learning models, would be crucial for further enhancing practical predictability of heatwaves in the follow-up work.

In the revised manuscript, we have incorporated the relevant discussions in lines 53-63 and lines 312-316:

Lines 53-63: “Operational forecasting systems can reliably issue warnings of heatwaves that last more than three days, with a lead time of two to three days [1]. Extending the lead time of heatwave forecasts requires an improved understanding of the background atmospheric circulation patterns associated with heatwaves [1, 2, 13, 14]. On the planetary scale, atmospheric high-pressure systems are dominated by persistent anomalous atmospheric circulation patterns, such as the quasi-stationary ridge or block in the atmospheric flow [9, 15], long-lived Rossby wave packets [9, 10, 12], or circumglobal resonant Rossby waves [16, 17]. However, current process-based numerical models are unable to reliably predict the locations and strengths of these relevant background patterns at the desired lead times [1, 18–22], which hinders both the prediction skill and our understanding of the mechanisms behind heatwaves[1, 22, 23]. ”

Lines 312-316: “To further enhance the ability to predict heatwaves in practical forecasting, in the future, a deeper understanding of RWPs is needed for anticipating the onset of propagating and standing patterns earlier, and integrating the link between propagation patterns and RWPs into the machine-learning or numerical forecast models [22, 23, 50]. ”

Typo errors:

1. Abstract “fix stations” should be “fixed stations”

Thanks. We have modified this typo in the revised manuscript.

2. Line 54 three days two days in advance

Thanks. We have improved this sentence in the revised manuscript, as below:

“Operational forecasting systems can reliably issue warnings of heatwaves that last for more than three days, with a lead time of two to three days [1]”

Reference

Ref. 1: Domeisen, D.I., Eltahir, E.A., Fischer, E.M., Knutti, R., Perkins-Kirkpatrick, S.E., Schar, C., Seneviratne, S.I., Weisheimer, A., Wernli, H.: Prediction and projection of heatwaves. *Nature Reviews Earth & Environment* 4(1), 36–50 (2023).

Ref. 2: Kornhuber, K., Lesk, C., Schleussner, C.F., Jagermeyr, J., Pfliegerer, P., Horton, R.M.: Risks of synchronized low yields are underestimated in climate and crop model projections. *Nature Communications* 14(1), 3528 (2023).

Ref. 3: Lin, H., Mo, R., Vitart, F.: The 2021 western north american heatwave and its subseasonal predictions. *Geophysical Research Letters* 49(6), 2021–097036 (2022).

Ref. 4: Lembo, V., Bordoni, S., Bevacqua, E., Domeisen, D. I., Franzke, C. L., Galfi, V. M., et al.: Dynamics, statistics, and predictability of Rossby waves, heat waves, and spatially compounding extreme events. *Bulletin of the American Meteorological Society*, 105(12), E2283-E2293 (2024).

Ref. 5: Tibaldi, S., Molteni, F.: Atmospheric blocking in observation and models. In *Oxford Research Encyclopedia of Climate Science* (2018).

Ref. 6: Schubert, S. and Lucarini, V.: Dynamical analysis of blocking events: spatial and temporal fluctuations of covariant Lyapunov vectors. *Quarterly Journal of the Royal Meteorological Society*, 142(698), pp.2143-2158 (2016).

Ref. 7: Chattopadhyay, A., Nabizadeh, E., Hassanzadeh, P.: Analog forecasting of extreme-causing weather patterns using deep learning. *Journal of Advances in Modeling Earth Systems*, 12(2), e2019MS001958 (2020).

Reviews on “Evidence for preferred propagating terrestrial heatwave pathways due to Rossby wave activity”

This work investigates the mechanisms underlying the spatial propagation of the terrestrial heat waves and find out four preferred propagation pathways of terrestrial heatwaves in the northern hemisphere. And each preferred pathways are consistent with the movement of Rossby wave trains. The detected propagation pathways are found to provide prior knowledge for occurrences of downstream heatwaves and thus can be used for identifying associated precursor signals. I find the results interesting and innovative, as they provide a compelling explanation for the occurrence of simultaneous extremely high temperatures across the globe within a short period. This insight sheds light on the underlying mechanisms that drive these widespread heatwave events. However, the method and expressions are a little hard to follow. Please see the detailed comments below.

Overall, the method parts are very complex and confusing,

- **Definition of Heatwaves:** The authors state in the manuscript, “*A heatwave with the spatial extent (around 300 km [26]) could be detected by multiple surrounding regions, and this is taken into account when analyzing heatwaves by the used complex network algorithm.*” However, it remains unclear how many hot regions a heatwave comprises or what the minimum spatial extent should be for an event to qualify as a heatwave. Could the authors clarify these points to ensure a consistent and precise definition of heatwaves in their analysis?
- **The calculation of ES measurements: I am confused about** the definition of μ , v , e and d . They are not clearly explained, which make the method hard to be understood. More specific, what exactly do the event time series e_i/e_j represent? The average temperature or maximum temperature or the start date/end date for a heatwave event?
- **Does the matrix A consist of modified ES values ?**
- **Local Searching algorithm:** how is the out-degree obtained?
- Also, I am confused about the expression of heatwaves in “region A or B” and “node A and B”.

The result descriptions:

- As stated in Line 142 “Inspections of all heatwaves over the source regions identified from Fig. 1 suggests that such propagating heatwaves have occurred many times over the past few decades.” Could you be more specific, or provide a global map for the frequency of heatwave occurrences.
- Figure 1 is interesting. It gives the four typical heatwave propagation pathways across the whole globe. The pathway over Asia indicates that heatwaves over the eastern coast regions of China generally originate from Ural. I am curious about where the heatwaves in southern China originate. Do they originate from the same source as the eastern part of China? Additionally, it would be helpful to clarify how much these four typical propagation pathways contribute to the overall heatwave pathways worldwide.
- How to obtain the centers of the 500-hPa pressure system and their uncertainty range in Figure 2?
- Line 186-187 The stationary heatwaves are firstly mentioned here but without a detail description. Please clarify how to distinguish the propagating and stationary heatwaves?
- How is R_{HO} calculated? Is it a distance or a ratio? So confused.
- How can we recognize the duration of blocking system from Fig. S19?

...